# Sensory experience controls dendritic structure and behavior by distinct pathways involving degenerins

**Sharon Inberg[1], Yael Iosilevskii[1], Alba Calatayud-Sanchez[2], Hagar Setty[3,4], Meital Oren-Suissa[3,4], Michael Krieg[2], Benjamin Podbilewicz[1]***

[1]Department of Biology, Technion-Israel Institute of Technology, Haifa, Israel; [2]ICFO - Institut de Ciencies Fotoniques, The Barcelona Institute of Science and Technology, Barcelona, Spain; [3]Department of Brain Sciences, Weizmann Institute of Science, Rehovot, Israel; [4]Department of Molecular Neuroscience, Weizmann Institute of Science, Rehovot, Israel

**\*For correspondence:**
podbilew@technion.ac.il

**Competing interest:** The authors declare that no competing interests exist.

**Abstract** Dendrites are crucial for receiving information into neurons. Sensory experience affects the structure of these tree-like neurites, which, it is assumed, modifies neuronal function, yet the evidence is scarce, and the mechanisms are unknown. To study whether sensory experience affects dendritic morphology, we use the *Caenorhabditis elegans'* arborized nociceptor PVD neurons, under natural mechanical stimulation induced by physical contacts between individuals. We found that mechanosensory signals induced by conspecifics and by glass beads affect the dendritic structure of the PVD. Moreover, developmentally isolated animals show a decrease in their ability to respond to harsh touch. The structural and behavioral plasticity following sensory deprivation are functionally independent of each other and are mediated by an array of evolutionarily conserved mechanosensory amiloride-sensitive epithelial sodium channels (degenerins). Calcium imaging of the PVD neurons in a micromechanical device revealed that controlled mechanical stimulation of the body wall produces similar calcium dynamics in both isolated and crowded animals. Our genetic results, supported by optogenetic, behavioral, and pharmacological evidence, suggest an activity-dependent homeostatic mechanism for dendritic structural plasticity, that in parallel controls escape response to noxious mechanosensory stimuli.

## Editor's evaluation

This is an important study on neuronal plasticity demonstrating that mechanosensory isolation of *C. elegans* nematodes induces homeostatic structural changes in the dendritic tree and differential response to mechanical stimulation. Convincing evidence is provided to show that both structural and behavioral outcomes are mediated by degenerins – a highly conserved family of ion channels. The study will be of broad interest to the neuroscience community.

## Introduction

The general structure of the nervous system has been known for over a century. Groundbreaking studies on synaptic plasticity and its underlying mechanisms have shown that before birth and in adult animals, neuronal activity is needed for synaptic remodeling (*Fox and Wong, 2005*; *Goodman and Shatz, 1993*; *Katz and Shatz, 1996*; *Wiesner et al., 2020*; *Zuo et al., 2005*). In contrast, the molecular mechanisms responsible for structural remodeling of dendritic trees, as a result of different

sensory inputs (experience), especially during adulthood, are less understood (*Kolb and Whishaw, 1998*; *Tavosanis, 2012*; *Wong and Ghosh, 2002*).

Mechanistic understanding of experience-dependent structural plasticity is primarily focused on activity sensation by calcium channels and *N*-methyl-D-aspartate (NMDA) receptors. These are known to induce downstream signaling cascades affecting, among others, the Rho family of small GTPases, calcium metabolism, and microtubule stability (*Ghiretti et al., 2014*; *Sin et al., 2002*; *Vaillant et al., 2002*; *Zhou et al., 2006*). Several neurological conditions including autism, Down syndrome, fragile X syndrome, and schizophrenia are characterized by abnormal dendritic spine structures (*Hu et al., 2020*; *Huebschman et al., 2020*; *Jan and Jan, 2010*; *Tendilla-Beltrán et al., 2019*). Uncovering the molecular basis of dendritic tree instability during development and adulthood, may shed light on neurological disease mechanisms and elucidate their behavioral phenotypes.

The dendrite morphology of the *Caenorhabditis elegans*' PVD bilateral neurons is composed of repetitive, stereotypical, and spatially organized structural units that resemble candelabra (*Figure 1A*), making it a useful platform to study dendritic morphogenesis in hermaphrodites and males (*Oren-Suissa et al., 2010*; *Iosilevskii et al., 2024*; reviewed in *Heiman and Bülow, 2024*). While some of the genetically programmed molecular mechanisms responsible for the morphogenesis and regeneration of PVD's dendritic trees are known (*Dong et al., 2013*; *Dong et al., 2015*; *Inberg et al., 2019*; *Kravtsov et al., 2017*; *Oren-Suissa et al., 2017*; *Oren-Suissa et al., 2010*; *Salzberg et al., 2013*; *Salzberg et al., 2014*; *Smith et al., 2010*; *Heiman and Bülow, 2024*), the influence of nurture (e.g. sensory experience) on its structure and function remain unexplored. The PVD mediates several sensory modalities (reviewed in *Goodman and Sengupta, 2019*), notably response to harsh mechanical stimuli (nociception) (*Chatzigeorgiou et al., 2010*), response to low temperatures (*Chatzigeorgiou et al., 2010*), and proprioception (*Albeg et al., 2011*; *Tao et al., 2019*, reviewed in *Krieg et al., 2022*). While the PVD response to low temperatures is mediated by transient receptor potential (TRP) channels (*Chatzigeorgiou et al., 2010*), nociception and proprioception are mediated by degenerins/epithelial sodium (Na$^+$) channels (DEG/ENaCs) expressed in the PVD (*Chatzigeorgiou et al., 2010*; *Husson et al., 2012*), which form homo- and hetero-trimers and are involved in force sensing. In mammals, some DEG/ENaCs such as ASIC1a participate in synaptic plasticity and cognitive functions such as learning and memory (*Baldin et al., 2020*; *Bianchi and Driscoll, 2002*; *Chen et al., 2015*; *Gillespie and Walker, 2001*; *Gobetto et al., 2021*; *Hill and Ben-Shahar, 2018*; *Mango and Nisticò, 2020*; *Welsh et al., 2002*). As sensory and social isolation affect the behavior and fitness of diverse animals (*Bailey and Moore, 2018*; *Kuhlman et al., 2014*; *Wilbrecht et al., 2010*; *Yu and Zuo, 2011*), including primates (*Harlow et al., 1965*), studying adult nematode somatosensory neurons can reveal possibly conserved mechanisms of dendritic plasticity, which are induced by sensory stimuli.

In gentle touch circuits, which are distinct from nociception (*Chalfie and Sulston, 1981*; *Li et al., 2011*; *Oren-Suissa et al., 2010*), *Rose et al., 2005* found that deprivation of mechanosensory stimulation generated by colliding conspecifics in the growing plate, resulted in modified glutamatergic signaling and reduced response to tap stimulation. Here, we adapted this mechanosensory stimulation paradigm (*Rose et al., 2005*), where the crowded worms are thought to represent the natural 'default' enriched mechanosensory state, to look into nociceptive circuits and identify the mechanism that couples mechanosensory experience to structural and functional dendritic plasticity (*Inberg et al., 2018*). We focused on how mechanosensory experience, perceived through DEG/ENaCs, affects structural plasticity of the PVD dendritic trees in adult *C. elegans*, and whether this entails behavioral consequences. We find that mechanosensory experience not only alters the PVD's dendritic structure in the adult, but also affects associated behavioral outputs. However, and in contrast to the prevalent hypothesis, these structural and behavioral properties are not correlated.

## Results

### Sensory isolation induces behavioral plasticity

*Rose et al., 2005* have shown that isolation of wildtype (WT) *C. elegans* causes a decreased response to gentle touch circuits when comparing to worms raised in a crowded setting. We utilized a similar paradigm to study nociception, a behavior associated with PVD neuron activation (*Oren-Suissa et al., 2010*; *Way and Chalfie, 1988*), using a behavioral assay that registers escape following prodding with a platinum wire (harsh touch assay; *Figure 1A*; *Chalfie and Sulston, 1981*; *Li et al., 2011*;

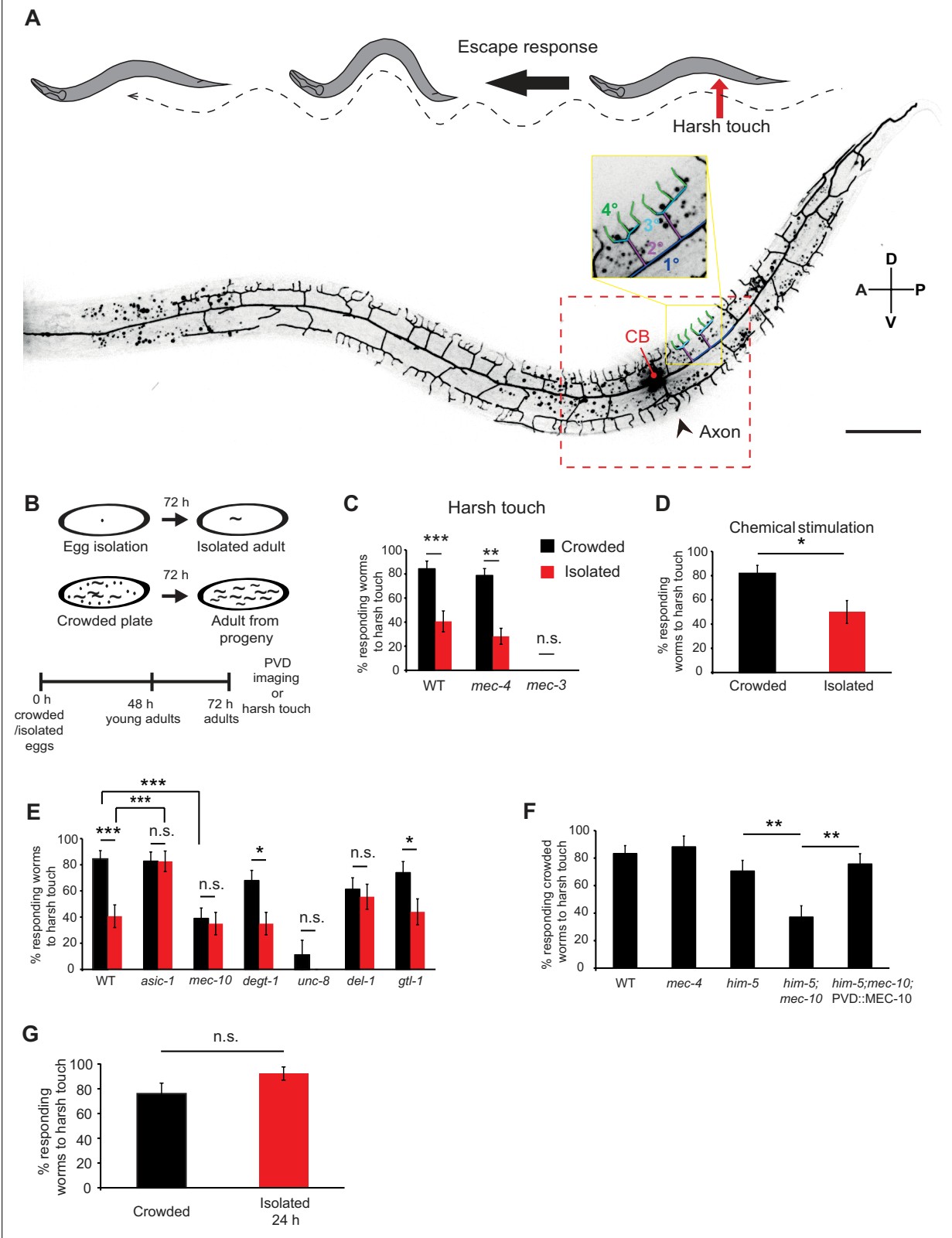

**Figure 1.** Mechanosensory deprivation during development reduces the behavioral response of the PVD neuron. (**A**) The PVD neuron dendritic tree, cell body (CB), and axon. The red arrow corresponds to the contact point with a platinum wire during posterior harsh touch, while the black arrow represents the behavioral escape response of the worm. The red dashed square represents the analyzed region around the CB. One representative candelabrum is colored by branch orders: blue primary (1°), purple secondary (2°), cyan tertiary (3°), and green quaternaries (4°). A – anterior; P – posterior; D – dorsal;

*Figure 1 continued on next page*

*Figure 1 continued*

V – ventral (scale bar, 50 μm). (**B**) Schematic of the isolation protocol followed at 72 hr by posterior harsh touch assay or PVD imaging of adult worms. (**C**) Isolation of embryos reduced the percentage of worms responding to harsh touch at adulthood. Crowded – black bars, Isolated – red bars. Wildtype (WT) N2 worms (Crowded, $n$ = 32; Isolated, $n$ = 32), *mec-4* (Crowded, $n$ = 52; Isolated, $n$ = 46), and *mec-3* (Crowded, $n$ = 12; Isolated, $n$ = 12). *mec-4* animals were assayed as adults after 96 hr to account for their slower rate of growth. (**D**) Growth in plates with chemical cues from adult hermaphrodites did not alter the reduced response rate of isolated *mec-4* animals to harsh touch (Crowded, $n$ = 33; Isolated, $n$ = 28). (**E**) Harsh touch response in crowded and isolated conditions for mutants of different DEG–ENaCs and the transient receptor potential (TRP) channel *gtl-1*. WT worms (same set of worms as in (C)). Crowded, $n$ = 32; Isolated, $n$ = 32, *asic-1* (Crowded, $n$ = 46; Isolated, $n$ = 30), *mec-10* (Crowded, $n$ = 38; Isolated, $n$ = 31), *degt-1* (Crowded, $n$ = 37; Isolated, $n$ = 31), *unc-8* (Crowded, $n$ = 18; Isolated, $n$ = 15), *del-1* (Crowded, $n$ = 31; Isolated, $n$ = 27), and *gtl-1* (Crowded, $n$ = 27; Isolated, $n$ = 25). (**F**) MEC-10 expression in the PVD rescues *mec-10* mutants' crowded-specific reduction in response to harsh touch. All strains contain the *ser2Prom3::Kaede* PVD marker construct, and were tested in the crowded conditions. WT, $n$ = 42; *mec-4*, $n$ = 17; *him-5*, $n$ = 34 (*him-5* was used as WT background for several strains after cross); *him-5; mec-10*, $n$ = 35; *him-5; mec-10; ser2Prom3::mec-10*, $n$ = 33. (**G**) Isolation for 24 hr in adulthood did not affect the response to harsh touch. Worms grown under crowded conditions were isolated for 24 hr as young adults and compared against their crowded age-matched counterparts in their response to harsh touch (Crowded, $n$ = 25; Isolated for 24 hr, $n$ = 26). The proportion of responding worms (percentage) ± standard error of proportion is shown. Fisher's exact test, *$p$ < 0.05, **$p$ < 0.01, ***$p$ < 0.001, n.s., not significant.

The online version of this article includes the following source data and figure supplement(s) for figure 1:

**Source data 1.** Original data file for *Figure 1* graphs on mechanosensory deprivation during development reduces the behavioral response of the PVD neuron.

**Figure supplement 1.** The reduction in response to harsh touch following isolation is PVD dependent and chemosensory independent.

**Figure supplement 1—source data 1.** Original data file for *Figure 1—figure supplement 1* on the reduction in response to harsh touch following isolation is PVD dependent and chemosensory independent.

*Oren-Suissa et al., 2010*). To study whether mechanosensory deprivation affects the nociceptive functions of the PVD, we isolated embryos into single plates where they grew 72 hr to adulthood and compared their behavioral response to harsh touch against same-aged adults that were grown on crowded plates (*Figure 1B*). We found that ~40% of isolated WT animals responded to harsh touch, compared with ~80% of animals grown in crowded plates (*Figure 1C*). To test whether gentle touch neurons are involved in this behavioral difference, we studied *mec-4(e1611)* mutants, in which gentle touch neurons are degenerated (*Caneo et al., 2019*; *Driscoll and Chalfie, 1991*; *Hedgecock et al., 1983*; *Li et al., 2011*; *Suzuki et al., 2003*), and obtained similar results (*Figure 1C*). Given that *mec-4(e1611)* exhibits the same experience-dependent behavioral plasticity as WT, we conclude that this phenomenon is independent from the *mec-4* gentle touch neurons. As a negative control, we used *mec-3(e1338)* mutants, that are harsh-touch-insensitive (*Way and Chalfie, 1988*) and found no responses for both groups (*Figure 1C*). Thus, isolation reduces the response to noxious stimuli, in a process that is independent of gentle touch circuits.

To determine whether the effect of isolation on nociception is related to the mechanosensory deficit itself (absence of channel gating to detect collisions; *Shi et al., 2016*; *Chatzigeorgiou et al., 2010*), in contrast to chemosensory stimuli, we used the *mec-4* strain to compare responses of isolated worms to responses of isolated worms grown in the presence of glass beads (resembling a method used by *Sawin et al., 2000*). Worms grown in isolation with beads had a similar response compared to animals grown in crowded plates (*Figure 1—figure supplement 1A*). While the beads are sufficient to increase isolated animals' responses, they do not fully recapitulate the crowded state (*Figure 1—figure supplement 1A*). To study whether the effect was chemosensory mediated, we tested plates pre-stimulated with pheromones as well as *osm-6* mutants (defective in olfactory sensory cilia; *Collet et al., 1998*). We found that isolation induced a reduction in the response to harsh touch, regardless of plate 'odor' (*Figure 1D*) or chemosensory function (*Figure 1—figure supplement 1B*). We note that the chemosensory defective *osm-6* mutants also have a defective ciliated mechanosensory PDE neuron (*Collet et al., 1998*) (which also mediates posterior harsh touch responses; *Li et al., 2011*). These results suggest that adult worms display a behavioral plasticity in PVD-related harsh touch circuits, which is dependent on mechanosensory experience, and is independent of the PDE neuron, olfactory function, and gentle touch neurons.

## MEC-10 and other mechanosensory channels mediate isolation-induced behavioral plasticity before adulthood

To study the genetic mechanisms behind the nociceptive response plasticity, we performed a candidate gene screen for DEG/ENaCs and TRP channels that are expressed in the PVD, where degenerins mediate mechanosensation and proprioception, while TRPs sense low temperatures. In all these signal transduction pathways some of the outcomes are behavioral responses (*Chatzigeorgiou et al., 2010*; *Huang and Chalfie, 1994*). We found that the WT-like isolation-induced reduction in harsh touch response was also present in *degt-1* DEG/ENaC and *gtl-1* TRP channel mutants, suggesting that these channels are not directly involved in behavioral plasticity following isolation (*Figure 1E*). In contrast, for *del-1*, *asic-1*, and *mec-10* DEG/ENaC mutants the difference between isolated and crowded conditions was undetectable, indicating that they are required for such behavioral plasticity. Interestingly, while the harsh touch response of *asic-1* mutants was consistently high, and similar to crowded WT worms, the response for *mec-10* mutants was consistently low, similar to isolated WT animals, regardless of sensory experience (*Figure 1E*). To test whether the response to harsh touch is dependent on DEG/ENaC activity, we used the DEG/ENaC blocker amiloride (*Ben-Shahar, 2011*; *Bianchi and Driscoll, 2002*) on WT worms that were grown in crowded plates. We found that the response to harsh touch was not affected by continuous growth in the presence of amiloride (*Figure 1—figure supplement 1C*). This result possibly supports the idea that different amiloride-sensitive epithelial sodium channels may have positive and negative effects on the response to harsh touch, which are masked by global inhibition (*Figure 1E*). This is further evident by the plasticity consequences of some double and triple DEG/ENaC mutant combinations, which are difficult to align into a coherent and simple epistatic genetic model (*Figure 4—figure supplement 1*).

The DEG/ENaC MEC-10 is expressed in the PVD and responds to mechanosensory signals (*Chatzigeorgiou et al., 2010*) including shear forces when expressed in heterologous cells (*Shi et al., 2016*; *Shi et al., 2018*). Since crowded *mec-10* mutants seem isolated-like in their behavioral response (*Figure 1E*), we asked how *mec-10* mediates the behavioral plasticity in crowded conditions. To test whether the activity of MEC-10 is required cell autonomously in the PVD, we expressed MEC-10 under a PVD-specific promoter in a *mec-10* mutant background (*Oren-Suissa et al., 2010*; *Tsalik et al., 2003*). We found that expression of MEC-10 in the PVD rescues the low response to harsh touch in crowded *mec-10* mutants, indicating that it acts cell autonomously to modulate behavioral plasticity (*Figure 1F*).

To determine whether isolation affects the response to harsh touch during development or in adults, we isolated crowded-raised young adults for 24 hr and found no difference in their response to harsh touch (*Figure 1G*). Thus, isolation-induced reduction in mechanosensation is determined before early adulthood, is PVD cell autonomous, and *mec-10* dependent.

## Experience affects PVD morphology via *mec-10* in adulthood

Altered sensory experience (such as stimuli deprivation) is known to drive synaptic plasticity in the nervous system (*Alvarez and Sabatini, 2007*; *Fox and Wong, 2005*; *Zuo et al., 2005*). However, little is known about the effects of mechanosensory deprivation on the architecture of sensory neurons. To study whether reduced mechanosensory experience can alter the dendritic structure of the PVD we followed the isolated growth paradigm (*Figure 1B*) and examined two morphological features of the PVD (*Figure 2A*): The fraction of ectopic (excessive, disordered) branches out of non-ectopic branches (those that form the 'ideal' candelabrum), and the percentage of straight fourth order (quaternary) branches, which form the terminal processes of the candelabrum.

When quantifying the branching pattern of the PVD in adulthood, WT (or WT-like *him-5* background) animals which were isolated as embryos showed an increase in ectopic branching compared with crowded age-matched worms (*Figure 2B*) and the quaternary branches assumed a more rippled shape (fewer straight 4ry branches, *Figure 2C*), regardless of the neuron marker utilized (see *Figure 2—figure supplement 1*).

To determine whether chemical stimulation plays a role in the observed morphological plasticity, animals were isolated onto pheromone-conditioned plates (*Maures et al., 2014*). Similar to the response to harsh touch (*Figure 1D*), chemical stimulation of the plates did not rescue the isolation-induced increase in ectopic branching or the decrease in straight quaternary branches (*Figure 2—figure supplement 2*). Additionally, we looked at *mec-4* gentle-touch-insensitive mutants (*Rose*

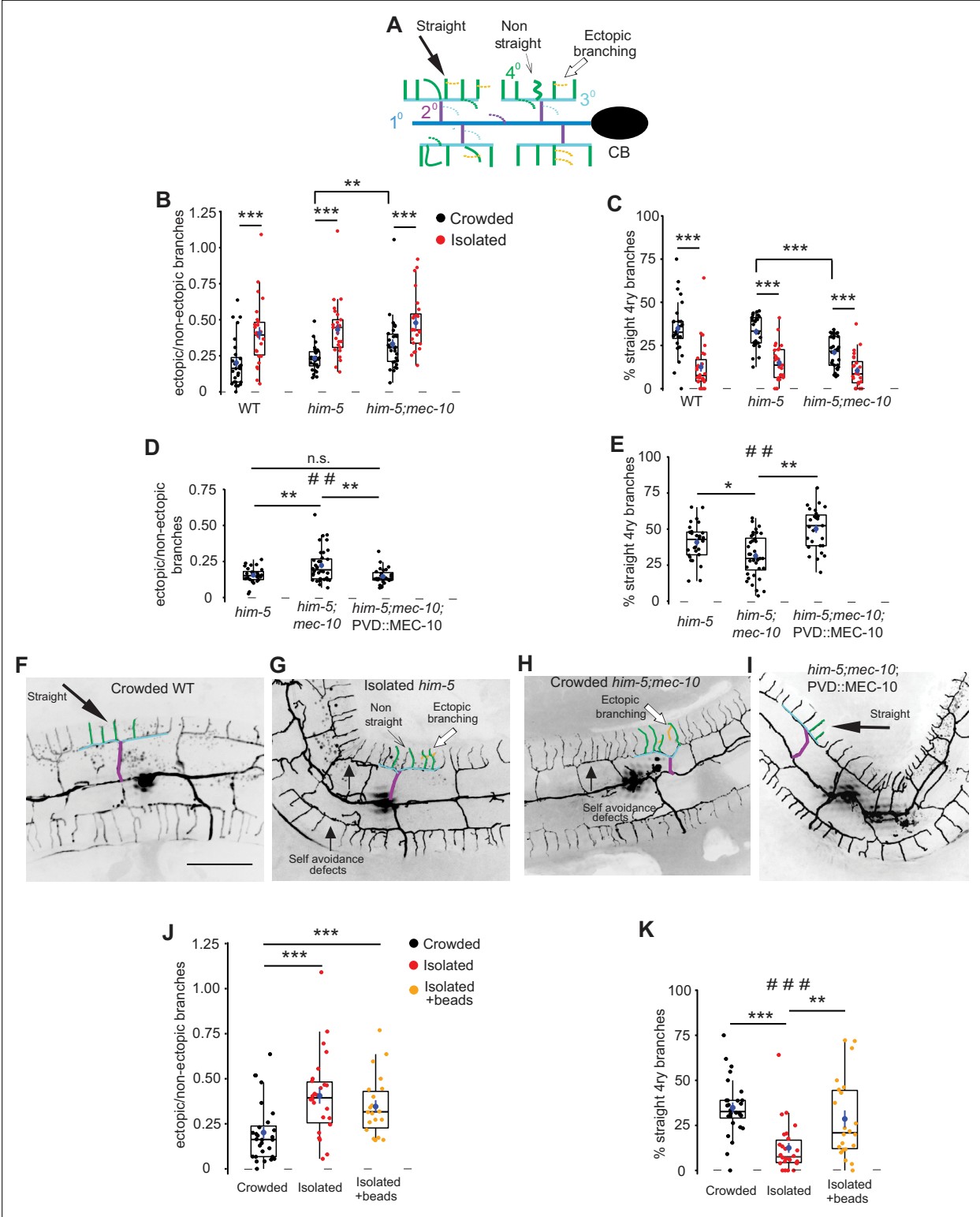

**Figure 2.** Mechanosensory deprivation and cell autonomous activity of *mec-10* affect the architecture of the PVD. (**A**) Schematic representation of the PVD dendritic structure, marking the morphological characteristics of interest: ectopic branches and quaternary branching geometry. Dashed lines represent ectopic branching at each order. Colors correspond to *Figure 1A*. (**B–G**) Both isolation and *mec-10* affect the structure of the PVD: (**B**) Isolation increases the fraction of ectopic branching, *mec-10* mutation increases ectopic branching in the crowded state. (**C**) Isolation decreases the percentage of straight quaternary branches, *mec-10* mutation decreases the percentage of straight quaternary branches in the crowded state. Crowded – black dots, isolated – red dots; wildtype (WT) (Crowded, *n* = 28; Isolated, *n* = 26), *him-5* (Crowded, *n* = 27; Isolated, *n* = 25; *him-5* was used as WT

*Figure 2 continued on next page*

*Figure 2 continued*

background for several strains after cross), *him-5; mec-10* (Crowded, *n* = 30; Isolated, *n* = 24). (**D**) Expression of *mec-10* in the PVD on the background of *him-5; mec-10* in the crowded state reduces ectopic branching and (**E**) increases the percentage of straight quaternary branches (Crowded *him-5*, *n* = 28; Crowded *him-5; mec-10*, *n* = 36; Crowded *him-5; mec-10*; PVD::MEC-10, *n* = 27). (**F–I**) Representative PVD images of WT, *mec-10*, and *mec-10*; PVD::*mec-10* in different growth conditions (scale bar, 50μm). (**J**) Embryo isolation with glass beads did not affect ectopic branching. (**K**) Embryo isolation with glass beads increased the percentage of straight quaternary branches (Crowded, *n* = 28; Isolated, *n* = 26; Isolated with beads, *n* = 22). Crowded and isolated WT are the same set of worms as in (B, C). Each dot represents a single worm. The mean ± SEM are shown in blue. Box plot with median and hinges for the first and third quartiles. The whiskers represent an estimated calculation of the 95% confidence interval. Kruskal–Wallis test, ##$p < 0.01$, ###$p < 0.001$, Mann–Whitney test with Bonferroni correction α = 0.0167. *$p < 0.05$, **$p < 0.01$, ***$p < 0.001$, n.s., not significant.

The online version of this article includes the following source data and figure supplement(s) for figure 2:

**Source data 1.** Original data file for *Figure 2* graphs on mechanosensory deprivation and cell autonomous activity of mec-10 affects the architecture of the PVD.

**Figure supplement 1.** The effect of isolation on the structure of the PVD is independent from the identity of the promoter driving expression of the reporter in the PVD.

**Figure supplement 1—source data 1.** Data for *Figure 2—figure supplement 1* on the effect of isolation on the structure of the PVD is independent from the promoter driving expression of the reporter in the PVD.

**Figure supplement 2.** The PVD undergoes isolation-dependent structural plasticity in the presence of pheromonal signals in the plate.

**Figure supplement 2—source data 1.** Original data for *Figure 2—figure supplement 2* on the PVD undergoes isolation-dependent structural plasticity in the presence of pheromonal signals in the plate.

**Figure supplement 3.** The effect of isolation on the structure of PVD is not mediated by the gentle touch mechanosensory neurons.

**Figure supplement 3—source data 1.** Original data for *Figure 2—figure supplement 3* graphs on the effect of isolation on the structure of PVD is not mediated by the gentle touch mechanosensory neurons.

**Figure supplement 4.** Isolation of eggs for 48 hr and adults for 24 hr is sufficient to induce changes in the structure of the PVD.

**Figure supplement 4—source data 1.** Original data for *Figure 2—figure supplement 4* graphs on isolation of eggs for 48 hr and adults for 24 hr is sufficient to induce changes in the structure of the PVD.

---

*et al., 2005*; *Suzuki et al., 2003*) and found that isolation caused a similar change to PVD structure as in the WT (*Figure 2—figure supplement 3*, compare with *Figure 2B, C*). In summary, these results suggest that mechanosensory experience controls morphological plasticity of PVD dendritic trees independently of *mec-4* and chemical stimulation.

We next sought to determine whether DEG/ENaCs mediate these experience-driven morphological alterations, by examining PVD morphology in crowded and isolated *mec-10* mutants. We found that crowded *mec-10* animals were different from WT crowded worms, and appear more isolated-like in terms of morphological features (more ectopic branches and fewer straight quaternary branches, compared to crowded *him-5* control animals; *Figure 2B, C*). These results suggest that PVD dendritic morphology is affected by sensory experience in a *mec-10*-dependent pathway. Importantly, these results confirm a role for the mechanosensory channel MEC-10, but not other degenerins (such as ASIC-1), in the plasticity of both the dendritic structure and the nociceptive function of the PVD in the crowded background.

To study whether *mec-10* acts cell autonomously to mediate morphological plasticity, we again utilized PVD-specific MEC-10 expression in a crowded *mec-10* mutant background and found it reduced the fraction of ectopic branches and increased the percentage of straight quaternary branches, compared to age-matched non-transgenic animals (*Figure 2D–I*). Thus, MEC-10 acts in the PVD to mediate morphological plasticity.

Having identified an isolation-induced morphological plasticity in adults (observed at 72 hr from embryo), we next sought to establish the temporal dependence of this effect, by comparing animals after only 48 hr, as young adults (see *Figure 2—figure supplement 4A*). Isolated worms showed a small but significant difference for the percentage of straight quaternary branches, but not in ectopic branching (*Figure 2—figure supplement 4B–E*) when compared with age-matched crowded worms, suggesting that isolation induces some morphological alterations during morphogenesis, while some relate to adult-stage maintenance of the neuron. We then isolated animals grown for 48 hr in crowded conditions, and found that isolation of young crowded adults for 24 hr induced changes in both measured parameters (*Figure 2—figure supplement 4F–I*), recapitulating isolation of embryos

(*Figure 2B, C*). Thus, isolation of adults for 24 hr is sufficient to affect PVD dendritic tree architecture, in contrast to the developmentally-established behavioral response (*Figure 1G*).

To further determine whether the morphological effect on PVD structure is solely mediated by mechanical cues, we used glass beads under isolated conditions. We found that while the presence of glass beads did not reduce the fraction of ectopic branching, it significantly increased the number of straight quaternary branches (compared with isolated animals without beads, *Figure 2J, K*). Thus, the mechanosensory stimuli of inert beads can partially rescue isolation-induced morphological changes to the PVD.

## Isolation triggers dynamic plasticity of the dendritic tree

To pinpoint the precise time interval required to mediate dendritic arborization changes in adulthood, we examined the effect of varying isolation times on the PVD morphology of crowded-raised young adults. We found that isolation of adults for 2 hr had no significant consequence on the structure of the PVD, while isolation of adults for 5 hr or longer significantly increased the proportion of ectopic branches and reduced the percentage of quaternary straight branches compared to crowded worms (*Figure 3A, B*).

To further study plasticity at the individual level, we compared the dendritic tree of individual worms grown under crowded conditions before and after 4 hr isolation. This revealed isolation induced more events of ectopic branch growth and fewer events of branch retraction (*Figure 3C*). Furthermore, this period was sufficient to induce an increase in ectopic branching and a decrease in percentage of straight quaternary branches (*Figure 3D, E*, *Figure 3—figure supplement 1*). Thus, 4 hr of isolation are sufficient to induce changes in the architecture of the adult PVD.

Since the geometry of quaternary branches shows rapid response to the amount of mechanosensory stimulation (*Figure 3A–E*), we next asked how dynamic is this experience-dependent morphological change. We performed a 3-hr time lapse imaging in crowded and isolated animals anesthetized with 1% tricaine (*Videos 1 and 2*). We found that while the number of straight quaternary branches in the isolated worms remained low and stable (~8%), for the crowded animals, we observed a gradual reduction in straight quaternary branches, starting after ~2 hr of isolation of individual worms on the slide (*Figure 3F, G*).

In summary, isolation-induced morphological plasticity is apparent within 2–3 hr from the switch from crowded conditions to growth in the absence of mechanical stimuli. This dynamic process involves simultaneous increase in ectopic branching and decrease in straight terminal branches.

## Activity affects morphology

It has been shown that activity and sodium influx via the DEG/ENaC UNC-8 promotes synapse elimination in *C. elegans* (*Miller-Fleming et al., 2016*). To test how global inhibition of DEG/ENaCs affects the morphology of the PVD neuron, we evaluated crowded worms that were grown on plates with amiloride compared to control worms. We found that blocking DEG/ENaCs by amiloride increased the fraction of ectopic branching and decreased the percentage of straight quaternary branches (*Figure 3—figure supplement 2A–D*). Thus, unlike harsh touch response, which is independent of sensory-level global inhibition (*Figure 1—figure supplement 1C*), global inhibition at the sensory level alters dendritic morphology. These results suggest that activity via degenerins modulates the structure of the PVD.

To test whether manipulations of other neuromodulators can affect the branching dynamics of the PVD, we utilized two anesthetics: tetramisole, which activates the nematode nicotinic acetylcholine receptors inducing muscle contraction (*Aceves et al., 1970*), and tricaine, which modulates neuronal activity by blocking sodium channels (*Katz et al., 2020*), and analyzed their effects during 2–3 hr of time lapse movies of crowded and isolated animals. We first tested the commonly used mixture of 0.1% tricaine and 0.01% tetramisole (the racemic mixture of the enantiomer levamisole) (*Kravtsov et al., 2017*) and found that it induces more growth than retraction of ectopic branches in crowded animals (*Figure 3—figure supplement 2E–G*). In contrast, 1% tricaine alone has the opposite effect – more retractions of ectopic branches (*Figure 3—figure supplement 2H-J*; *Videos 1 and 2*). In addition, 1% tricaine induces more growth for isolated compared to crowded worms, indicating that there are activity-dependent intrinsic factors that are different for the two experience states. These results suggest that modulating neuronal activity triggers structural modifications in the PVD, indicating a

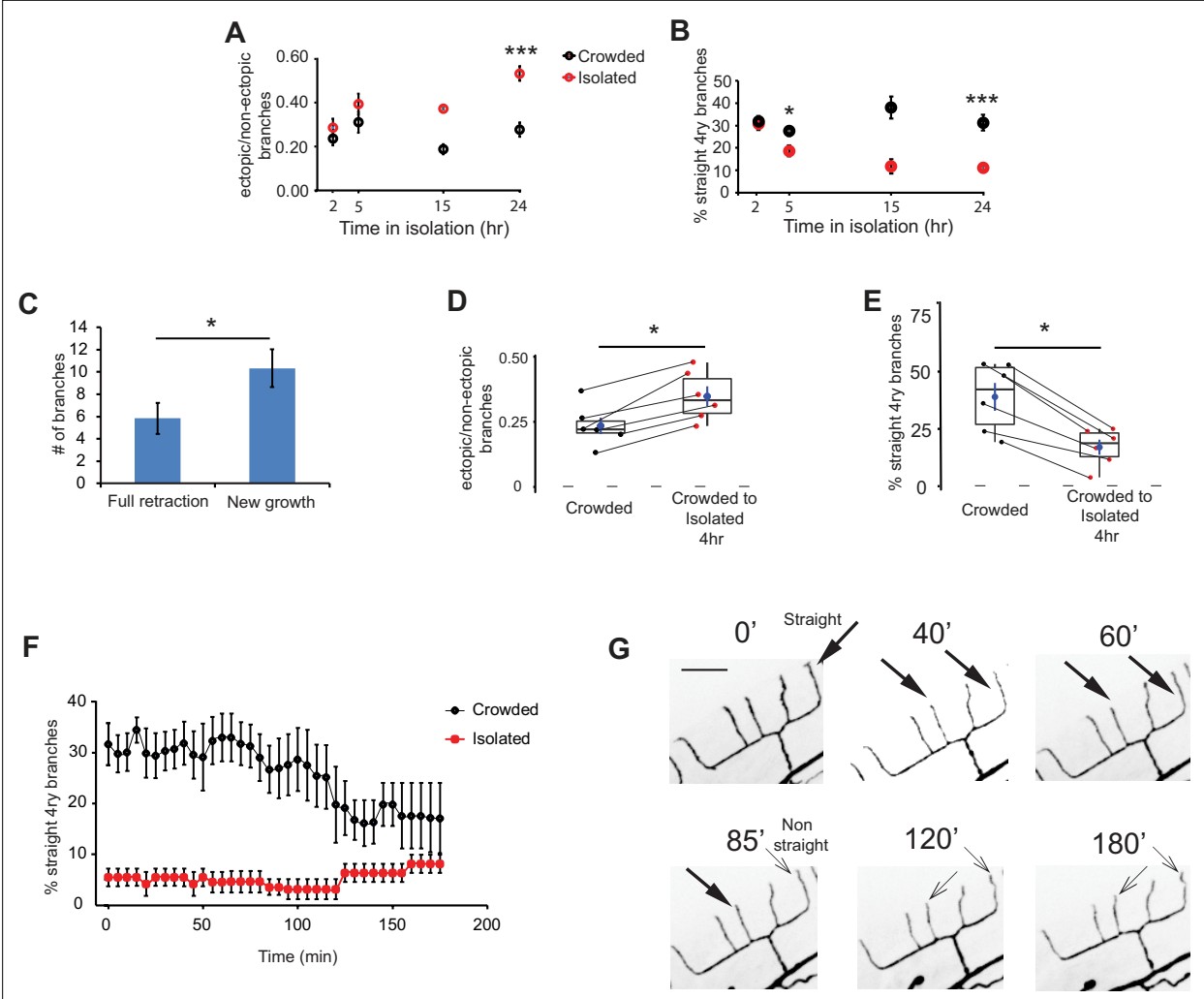

**Figure 3.** Adult isolation for less than 4 hr affects the structure of the PVD. (**A**) Isolation of crowded worms induces a time-dependent increase in the proportion of ectopic branches, and (**B**) a decrease in the number of straight quaternary branches (2 hr: Crowded, n = 24, Isolated, n = 22; 5 hr: Crowded, n = 24, Isolated, n = 27; 15 hr: Crowded, n = 3, Isolated, n = 3; 24 hr: Crowded, n = 20, Isolated, n = 21; animals imaged using polystyrene beads for immobilization, see Materials and methods). (**C**) Isolation for 4 hr induces more growth of new branches than retraction of existing ones. Ectopic branch dynamics were compared for crowded animals against their state after 4 hr isolation. Imaging used 1% tricaine, see Materials and methods. (**D**) Isolation of crowded worms for 4 hr increases the ratio between ectopic/non-ectopic branches in individual animals and (**E**) reduces the percentage of straight quaternary branches (n = 6). (**F**) Crowded, but not isolated worms, show dynamic reduction in the percentage of straight quaternary branches during a 3-hr time lapse movie in 1% tricaine; see *Videos 1 and 2* (Crowded, n = 5; Isolated, n = 6). (**G**) Representative time lapse frames of a crowded worm in 1% tricaine. Thick and thin arrows represent straight and non-straight quaternary branches, respectively (scale bar, 25 μm). For (**D, E**), the lines connecting two points refer to the same crowded specimen at time 0 and after 4 hr isolation. Box plot representation as in *Figure 2*. For (**A, B**) Mann–Whitney test, *p < 0.05, ***p < 0.001. For (**C–E**), Wilcoxon test (for two related samples), *p < 0.05. The mean ± SEM are shown for panels A–C and F.

The online version of this article includes the following source data and figure supplement(s) for figure 3:

**Source data 1.** Original data file for *Figure 3* graphs on adult isolation for less than 4 hr affects the structure of the PVD.

**Figure supplement 1.** Four hours are sufficient to induce an increase in ectopic branching and a decrease in percentage of straight branches.

**Figure supplement 2.** Activity-modulating pharmacological agents affect the structure and the dynamics of PVD branch growth and retraction.

**Figure supplement 2—source data 1.** Original data for *Figure 3—figure supplement 2* graphs on activity-modulating pharmacological agents affect the structure and the dynamics of PVD branch growth and retraction.

possible structural sensitivity to the amount of neuronal activity. In summary, we found that mechanosensory experience, DEG/ENaC protein presence and activity, and pharmacological targeting of activity dynamically affect the structure of the PVD.

## PVD structure and function are independent

After establishing that mechanosensory experience induces both a behavioral and a structural plasticity of the PVD, we asked whether there is a causal link between the morphology of the dendritic tree of the PVD and its function as a nociceptor (*Hall and Treinin, 2011*). We followed the isolation protocol described in *Figure 1B* for seven combinations of DEG/ENaC mutants and analyzed their response to harsh touch (*Figure 4—figure supplement 1*) and their PVD structure (*Figure 4—figure supplement 2*). In order to compare the morphological properties of the different DEG/ENaC genotypes under different sensory experience (crowded, isolated; *Figure 4—figure supplement 2*), we used discriminant analysis. This provides a supervised classification method to combine all the analyzed morphological phenotypes, including loss of self-avoidance between adjacent candelabra (*Figure 4—figure supplement 3*), into a certain value characterizing the PVD morphological state (*Figure 4A*). We then superimposed the behavioral results of the DEG/ENaC mutants' response to harsh touch (*Figure 4—figure supplement 1*) on the morphological clustering, as a binary-like property (<45% responding for isolated WT vs. >65% for crowded WT, as shown in *Figure 4—figure supplement 1*). We found no correlation between the morphology of the PVD and the response to harsh touch when testing the different combinations of genotypes and treatments. For example, isolated *mec-10; degt-1* double mutants show crowded-like morphology with isolated-like behavioral response (*Figure 4A*, *Figure 4—figure supplement 1*, *Figure 4—figure supplement 2*). These findings suggest that while degenerins are required for isolation-induced plasticity of both traits, response to harsh touch is independent of the structural alteration of the PVD.

An additional line of evidence supporting the independence of the behavioral and morphological phenotypes was demonstrated by isolation of young adult worms for 24 hr (*Figure 3*). This isolation affects the structure of the PVD (*Figure 2—figure supplement 4*) but has no effect on the response to harsh touch (*Figure 1G*). Amiloride also has no effect on harsh touch response (*Figure 1—figure supplement 1C*) but does alter dendritic morphology (*Figure 3—figure supplement 2A–D*). Finally, to directly demonstrate that these two features are independent, we assayed harsh touch responses of isolated animals and then analyzed each individual animal for its PVD morphology. We then compared the dendritic morphology of responding and non-responding worms, and found that the morphological parameters were similar (*Figure 4B–E*). Thus, analysis at the level of individual worms failed to

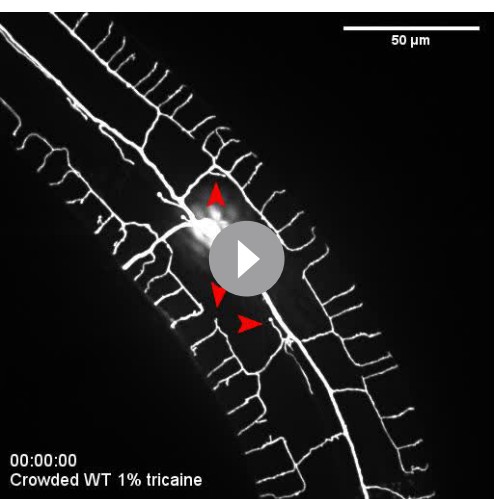

**Video 1.** Time lapse of cell body and arborizations of a wildtype young adult grown under crowded conditions and anesthetized with 1% tricaine. Red arrows point to retracting branches. See *Figure 3—figure supplement 2I* for more details.

https://elifesciences.org/articles/83973/figures#video1

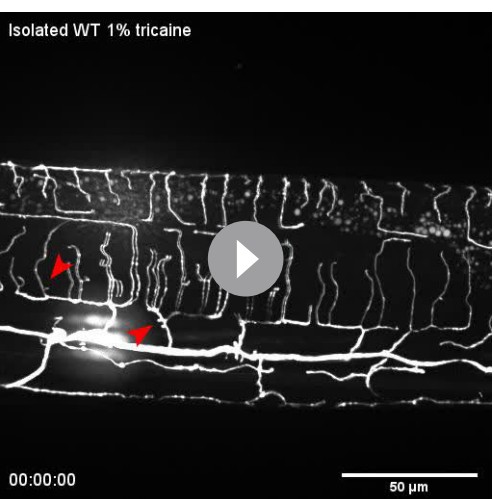

**Video 2.** Time lapse of cell body and arborizations of a wildtype young adult grown in isolated conditions and anesthetized with 1% tricaine. Red arrows point to retracting branches. See *Figure 3—figure supplement 2J* for more details.

https://elifesciences.org/articles/83973/figures#video2

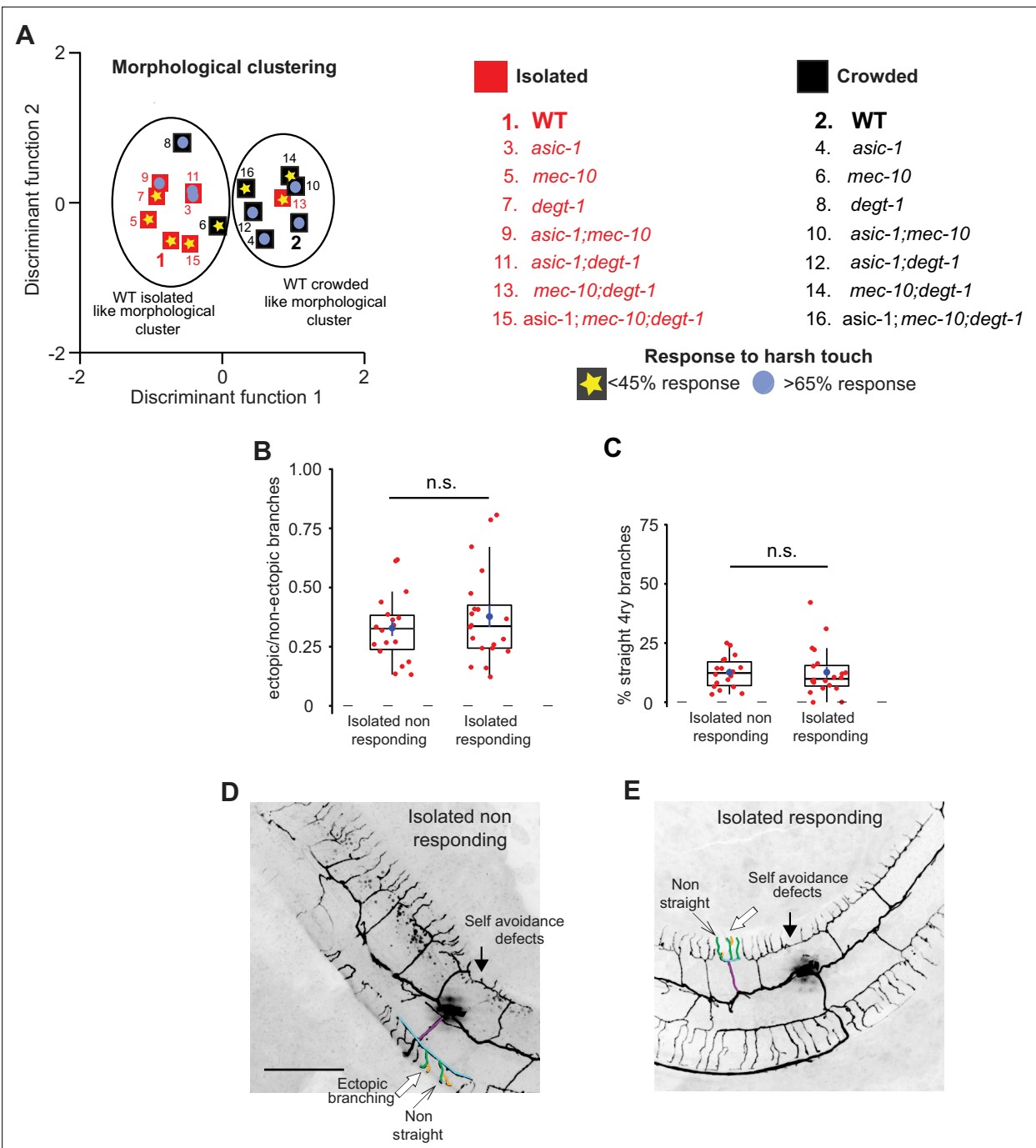

**Figure 4.** Response to harsh touch and PVD morphology are independent. (**A**) Discriminant analysis shows independence of harsh touch response from the PVD's morphological classification. Squares indicate the centroids for morphological characteristics analyzed in *Figure 4—figure supplement 2A–C*. The response to harsh touch (*Figure 4—figure supplement 1*) is illustrated by its magnitude (low, <45% in yellow star; high, >65% in light blue circles). The different genotypes are numbered in the list on the right. Crowded – black, Isolated – red. (**B–E**) Isolated animals show similar PVD morphology regardless of their touch response: (**B**) isolated responding and non-responding animals are not different with regard to the fraction of ectopic branching, and (**C**) the geometry of quaternary branches (Isolated non-responding, *n* = 18; Isolated responding, *n* = 20). Box plot representation as in *Figure 2*. Mann–Whitney test. n.s., not significant. (**D, E**) Representative PVD images of responding and non-responding worms (scale bar, 50 μm).

The online version of this article includes the following source data and figure supplement(s) for figure 4:

**Source data 1.** Original data file for *Figure 4* graphs on response to harsh touch and PVD morphology is independent.

**Figure supplement 1.** The DEG/ENaC *asic-1* and *mec-10* mediate experience-dependent behavioral plasticity following isolation.

*Figure 4 continued on next page*

*Figure 4 continued*

**Figure supplement 1—source data 1.** Original data for *Figure 4—figure supplement 1* on the DEG/ENaC *asic-1* and *mec-10* mediate experience-dependent behavioral plasticity following isolation.

**Figure supplement 2.** The DEG–ENaCs, *mec-10* and *degt-1*, mediate mechanosensory-dependent structural changes in the PVD.

**Figure supplement 2—source data 1.** Original data for *Figure 4—figure supplement 2* on the DEG–ENaCs, *mec-10* and *degt-1*, mediate mechanosensory-dependent structural changes in the PVD.

**Figure supplement 3.** Isolation induces increase in loss of self-avoidance defects, in a *mec-10*-dependent manner.

**Figure supplement 3—source data 1.** Original data for *Figure 4—figure supplement 3* on isolation induces increase in loss of self-avoidance defects, in a *mec-10*-dependent manner.

demonstrate a correlation between the morphology and the response to harsh touch. In summary, the morphological and behavioral phenotypes were independently affected by sensory experience via degenerins. We cannot exclude the possibility that other functions of the PVD, like the response to low temperatures (*Chatzigeorgiou et al., 2010*) and proprioception (*Albeg et al., 2011*) are more tightly associated to the structure of the PVD, nor that the morphological changes induced by isolation are too minor to constitute a difference in neuronal function.

In addition to the isolation-induced changes in the number of ectopic branches and the percentage of straight quaternary branches, we found that worms raised in isolation are also more likely to lose the self-avoidance between two adjacent menorahs (candelabra). This effect is also *mec-10* dependent, but appears to act cell non-autonomously. In addition, it is chemosensory and amiloride independent (*Figure 4—figure supplement 3*).

## MEC-10 and DEGT-1 localization is experience dependent

Differential localization of degenerins can affect both the behavioral response to harsh touch and the structural properties of the neuron. When considering the effect of isolation, we hypothesized that changes in the localization patterns of DEG/ENaC can account for plasticity at both the behavioral (*Figure 1*) and the structural level (*Figure 2*). Since MEC-10 and DEGT-1 tend to co-localize within the PVD (*Chatzigeorgiou et al., 2010*), we analyzed the interaction between these two proteins, under different mechanosensory experiences. We found that MEC-10 localization in the plasma membrane, in intracellular vesicular compartments of the axon and in the quaternary branches was reduced after isolation (*Figure 5A–D*, *Videos 3 and 4*). In contrast to MEC-10 (*Figure 5A–D*), DEGT-1 localization is reduced only in the cell body following isolation (*Figure 5E–H*). Furthermore, *degt-1* mutants show reduced MEC-10 signal, and more importantly, abrogated the isolation-induced reduction in MEC-10 localization at the quaternary branches and the axon (*Figure 5A–D*). In the reciprocal experiment, DEGT-1 localization was affected in *mec-10* mutants, as isolated worms exhibit increased localization to the cell body compared with WT isolated worms (*Figure 5E–H*). Thus, mechanosensory experience also induces plasticity in the localization pattern of MEC-10 and DEGT-1. We propose this differential localization may be part of the mechanism that independently and locally modulates dendritic and axonal properties, to affect both the structure and the function of the PVD, respectively.

## Optogenetic stimulation suggests behavioral plasticity is post-sensory

We have shown that sensory deprivation (isolation) affects the localization of two different mechanoreceptors, the degenerins MEC-10 and DEGT-1, in the dendrites, axon and soma of the PVD. It is conceivable that the differential localization of degenerins in different domains of the PVD may affect the morphology and function of the PVD. Thus, we decided to activate the PVD independently of the endogenous mechanoreceptors and study whether the escape behavior is similar in isolated and crowded animals. We hypothesized that if we circumvent the normal degenerin-mediated mechanostimulation, exciting the PVD downstream to sensory activity, the animals grown in isolation will respond to the same degree as if they were in crowded conditions and independently of the dendritic tree morphology. Thus, we used optogenetic stimulation with Channelrhodopsin (ChR2) expressed in the PVD (*Husson et al., 2012*) to activate the neuron while bypassing sensory perception. We found that isolation significantly reduced the percentage of worms responding to optogenetic stimulation of the PVD (*Figure 5I*), indicating that the plasticity in the response is acting downstream to PVD

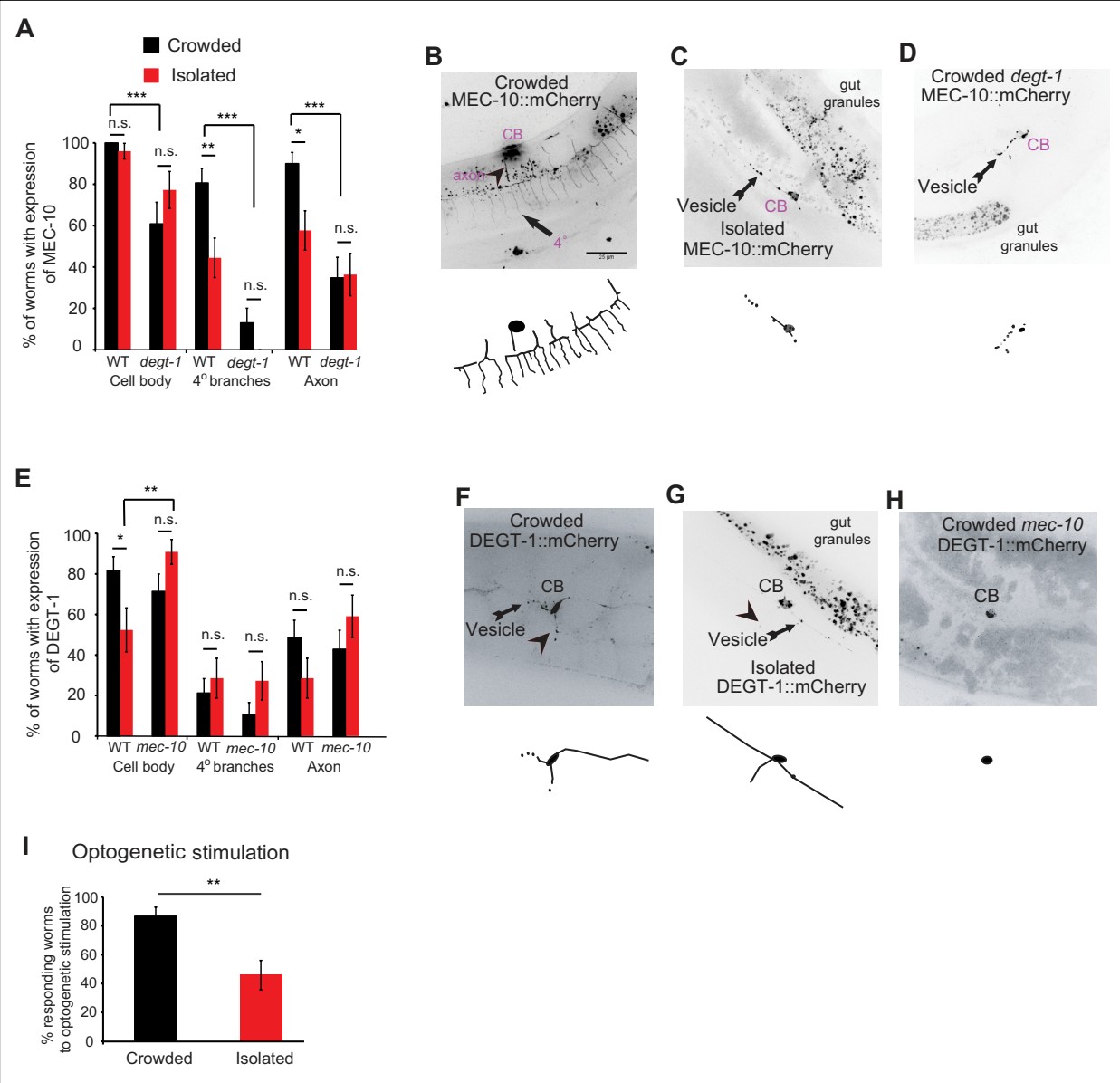

**Figure 5.** Mechanosensory-dependent localization of degenerins and optogenetics. (**A**) PVD::MEC-10::mCherry localization (shown and labeled in purple, and in blue in *Figure 7*) is reduced in the quaternary branches and the axon following isolation, in a *degt-1*-dependent manner (wildtype [WT]: Crowded, *n* = 31; Isolated, *n* = 27; *degt-1* mutants: Crowded, *n* = 23; Isolated, *n* = 22). (**B–D**) Representative images and reconstructions for PVD::*mec-10*::mCherry localization for crowded and isolated WT worms, and crowded *degt-1* worms (scale bar, 25 µm). (**E**) PVD::*degt-1*::mCherry localization level is reduced at the cell body, but not in the quaternary branches or the axon, following isolation, in a *mec*-10-dependent manner (WT: Crowded, *n* = 33; Isolated, *n* = 21; *mec-10* mutants: Crowded, *n* = 28; Isolated, *n* = 22). The percentage of expressing worms ± standard error of proportion is shown. (**F–H**) Representative images and reconstructions for PVD::*degt-1*::mCherry localization for crowded and isolated WT worms and crowded *mec-10* worms. (**I**) Isolation leads to a reduced escape response following optogenetic photoactivation of Channelrhodopsin 2 in the PVD (Crowded, *n* = 30; Isolated, *n* = 24, grown on All Trans Retinal. No response was observed for worms grown without All Trans Retinal). Fisher's exact test, *p < 0.05, **p < 0.01, ***p < 0.01, n.s., not significant.

The online version of this article includes the following source data and figure supplement(s) for figure 5:

**Source data 1.** Original data file for *Figure 5* graphs on mechanosensory-dependent localization of degenerins and optogenetics.

**Figure supplement 1.** Optogenetic stimulation of isolated worms does not affect the morphology of the PVD.

**Figure supplement 1—source data 1.** Original data file for *Figure 5—figure supplement 1* on optogenetic stimulation of isolated worms does not affect the morphology of the PVD.

activation, mechanosensory channels and signal transduction pathways. This activation is sensitive to isolation probably because it is acting pre- or postsynaptically.

To determine whether optogenetic stimulation of isolated animals can reverse the morphological changes induced by the absence of mechanical stimuli in PVD, we used optogenetic stimulation of the PVD and found no significant difference between isolated and isolated optogenetically stimulated animals in any of the measured PVD structural characteristic (*Figure 5—figure supplement 1*). Thus, optogenetic stimulation on isolated animals is not sufficient to convert their dendritic trees to crowded-like. In addition, the reduced response of isolated animals to optogenetic stimulation suggests the escape response is not dependent on the structure of dendritic trees but on unknown downstream pathways.

## Calcium dynamics in response to harsh touch are similar in isolated and crowded animals

Having shown that PVD optogenetic activation elicits a weaker escape response in isolated animals, we asked whether this occurs at the level of PVD itself, or originates further downstream.

To determine whether spontaneous $Ca^{++}$ dynamics in PVD is different between crowded and isolated animals we first observed the baseline calcium dynamics in animals grown isolated and crowded. We found no significant differences in the calcium levels between the isolated and crowded animals (*Figure 6—figure supplement 1*). To directly elicit PVD mechanosensory activation in isolated and crowded animals, we performed calcium recordings under mechanical stimuli using a custom microfluidic device (*Nekimken et al., 2017*). The device allows for the controlled application of mechanical stimulation, and was previously applied to study gentle touch receptor neurons (TRNs) (*Sanfeliu-Cerdán et al., 2023*) and AVG (*Setty et al., 2022*). Similar devices were used for PVD (*Cho et al., 2017*; *Tao et al., 2019*). To emulate harsh touch using our device, a 3-bar pressure caused a 15-µm cuticle displacement, greater than the indentation used for studying TRNs and similar to the tail stimulation paradigm used by Setty and colleagues (*Setty et al., 2022*). Previously, we have shown that TRNs responded most efficiently to a high frequency stimulation but were largely insensitive to stimuli delivered below 2 Hz (*Nekimken et al., 2017*). We applied three different stimulus profiles: a single step indentation (step), similar to a brief manual harsh touch prod, as well as a gradual pressure 'ramp' increase ('sawtooth'; in which the pressure increases slowly to 3 bar within 2 s) and a 10-Hz 'buzz' vibration (see Materials and methods). Even though we observed a strong and robust activation of PVD for all three stimuli tested (*Figure 6*),

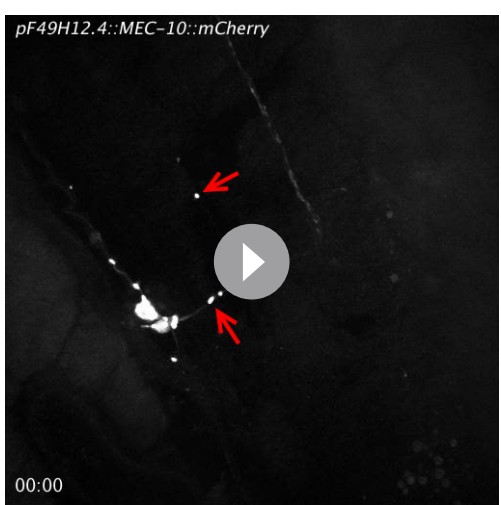

**Video 3.** Localization pattern of MEC-10::mCherry in the PVD crowded adult. MEC-10 is localized in moving vesicles, indicated by red arrow. Six z-stack series (~60 optical slices for each) were taken around the cell body every 3 min.

https://elifesciences.org/articles/83973/figures#video3

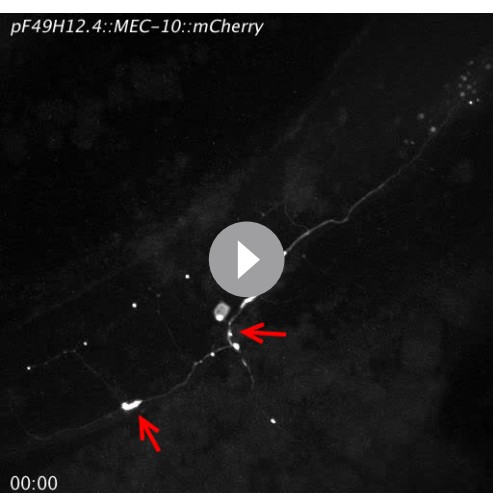

**Video 4.** Localization pattern of MEC-10::mCherry in the PVD for crowded adult. MEC-10 is localized in moving vesicles, indicated by red arrow. At time 00:00 quaternary branches can be seen. Six z-stack series (~60 optical slices for each) were taken around the cell body every 2 min.

https://elifesciences.org/articles/83973/figures#video4

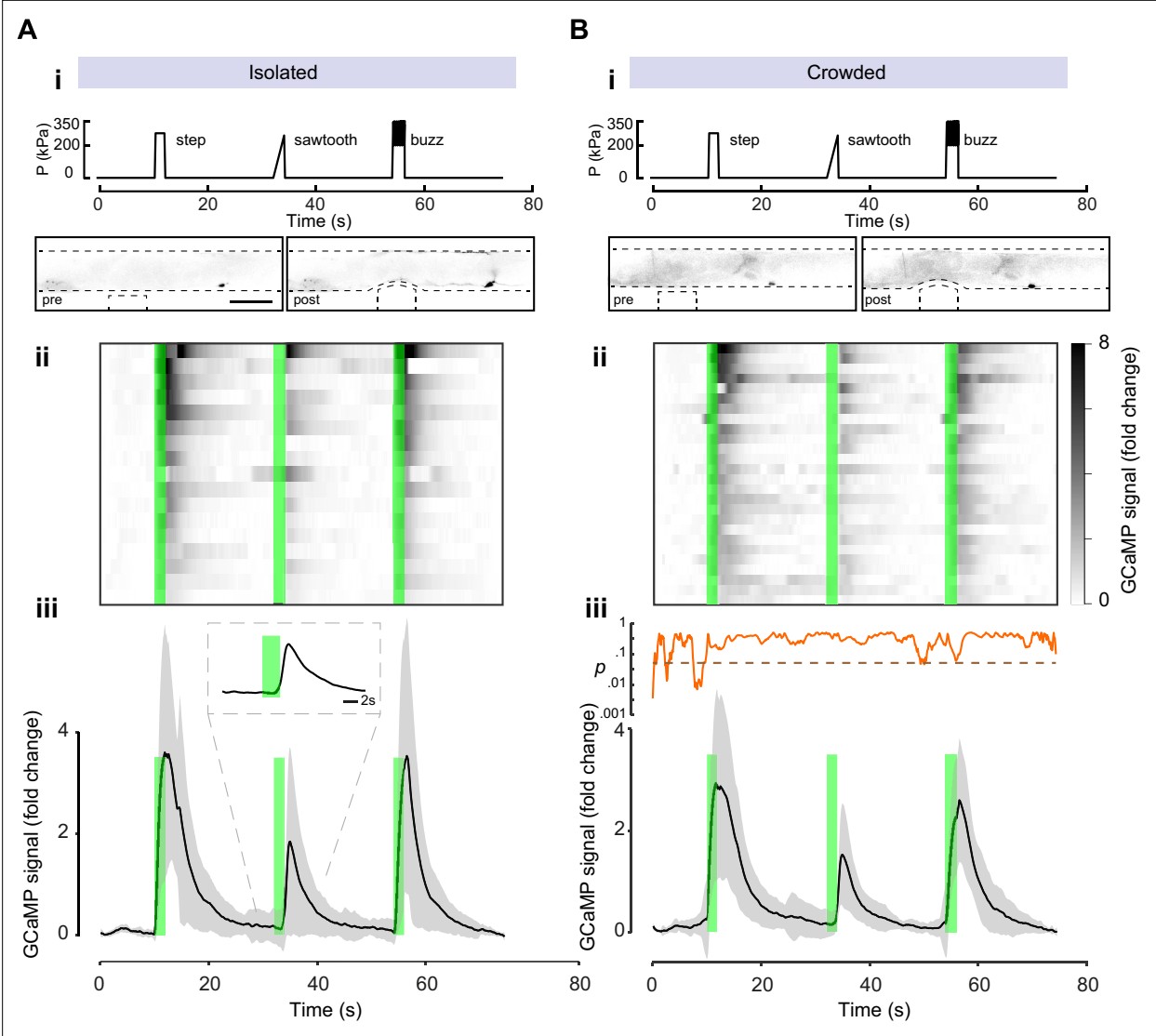

**Figure 6.** The calcium response to mechanical stimulation shows no significant difference between crowded and isolated worms. Mechanically induced calcium recordings of PVD of animals grown as (**A**) isolated individuals or (**B**) in crowded conditions. (**i**) Stimulus profile displaying a single step, sawtooth indentation or 10 Hz `buzz'. Bottom panels show examples of negative images from experiments at pre- and post-stimulation; note PVDs' cell bodies about 200 μm to the right of the location of the mechanical stimuli. Scale bar, 50 μm. (**ii**) Kymograph of individual response for $N = 17$ animals and $N = 27$ animals for the crowded conditions. Green bars, time of application of a mechanical stimulus. (**iii**) Average GCaMP6s signal derived from the cell body of PVD for the duration of the recordings, black line. $N$ = number of recordings. Mean ± standard deviation, gray region. Inset in panel iii shows the running p-value, comparing the crowded and isolated conditions (dashed line indicates alpha level of significance ∝ = 0.05).

The online version of this article includes the following source data and figure supplement(s) for figure 6:

**Source data 1.** Original data file for *Figure 6* graphs on calcium response to mechanical stimulation.

**Figure supplement 1.** Basal calcium dynamics are indistinguishable for crowded and isolated worms.

**Figure supplement 1—source data 1.** Original data file for *Figure 6—figure supplement 1* on basal calcium dynamics is indistinguishable for crowded and isolated worms.

**Figure supplement 2.** Calcium response in the PVD is sensitive to both onset and offset of the mechanical stimulus signals.

**Figure supplement 2—source data 1.** Original data file for *Figure 6—figure supplement 2* on calcium response in the PVD is sensitive to both onset and offset of the mechanical stimulus signals.

the response to the 'sawtooth' was significantly weaker (*Figure 6A*). Moreover, we noticed that the response correlated with the abrupt offset of the force in the 'saw tooth', rather than the gradual onset (*Figure 6Aiii*). To corroborate this result, we also tested a 20-s square step, and indeed found a separate response at the onset as well as the offset (*Figure 6—figure supplement 2*, Video 7). Thus, PVD, like TRNs, responds better to faster strain rates and adapts under continuous deformation (*Eastwood et al., 2015*). Using this analysis, we found that both isolated and crowded animals responded similarly and efficiently across all three types of stimuli (*Figure 6*, *Videos 5 and 6*). Together, these results suggest that the difference in the behavioral response to external mechanical stimuli is due to a pre- and/or postsynaptic effect following mechanosensory deprivation and not a direct consequence of the morphological changes in the structure of the PVD or its intrinsic activity at the level of the cell body.

In summary, our genetic and pharmacological evidence suggest that dendritic structural plasticity is an autonomous activity-dependent homeostatic mechanism. Combined with optogenetic testing and calcium imaging for mechanically stimulated PVD, our results indicate that dendritic tree plasticity is independent from downstream processes that affect escape behavior in response to harsh touch.

## Discussion

From the evolutionary point of view, dendritic trees, their morphogenesis (*Heiman and Bülow, 2024*) and their structural complexity remain mysterious objects, despite many efforts to understand the contribution of arborization complexity to dendritic physiology and function (*Häusser and Mel, 2003*).

Previous research has demonstrated both cell autonomous (*Aguirre-Chen et al., 2011*; *Oren-Suissa et al., 2010*; *Salzberg et al., 2014*) and cell non-autonomous (*Dong et al., 2013*; *Salzberg et al., 2013*) mechanisms that regulate the PVD's dendritic morphogenesis during development. Some studies have also focused on regeneration and aging effects on the tree structure of the PVD, revealing plasticity in the adult stage (*Iosilevskii and Podbilewicz, 2021*; *Kravtsov et al., 2017*; *Oren-Suissa et al., 2017*), similar to what has been shown for *Drosophila* sensory neurons (*DeVault et al., 2018*). We found, by using mutants for DEG/ENaCs, chemosensory stimulation, pharmacology, optogenetic stimulation, and glass beads, that the dendritic structure of the PVD and the behavioral response to harsh touch are activity- and mechanosensory dependent, but appear to be chemosensory independent. It is still conceivable that chemosensation can affect nociception and proprioception, in a cross modal plasticity mechanism, but at the moment we do not have evidence for this scenario. In contrast, we show that DEG/ENaCs mechanosensory channels (degenerins) affect the architecture of the PVD.

Our results suggest that 'nurture', manifested as mechanosensory experience, activates mechanotransduction signaling, via DEG/ENaCs amiloride-sensitive activity, to stabilize the homeostatic structure of the dendritic tree in adults. The sensory experience induced when one body side of the worm is in contact with the agar plate is not sufficient for this purpose (*Figure 7—figure supplement 1*); however, other worms on the plate, or the presence of glass beads, elicit significant structural

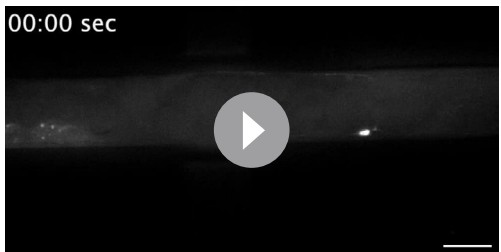

**Video 5.** Calcium dynamics in the PVD of an isolated worm after mechanosensory stimulation. PVD cell body received ipsilateral application of a mechanical stimulus, the calcium responses from the PVD were measured. See *Figure 6* for more details. Scale bar, 40 μm.

https://elifesciences.org/articles/83973/figures#video5

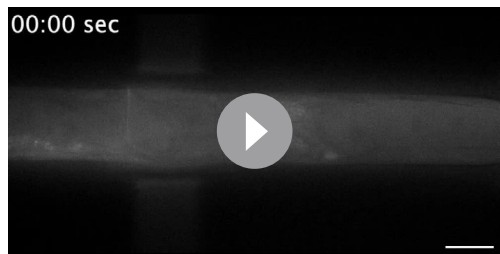

**Video 6.** Calcium dynamics in the PVD of a representative crowded worm after mechanosensory stimulation. PVD cell body received ipsilateral application of a mechanical stimulus, the calcium responses from the PVD were measured. See *Figure 6* for more details. Scale bar, 40 μm.

https://elifesciences.org/articles/83973/figures#video6

alterations. Moreover, the cell autonomous activity of MEC-10 in the PVD was found to be both necessary and sufficient to preserve the crowded-experience phenotype in terms of the simplified structure of the PVD. While there are other multiple plausible explanations, our failure in remodeling the isolated-state arbor by repeated optogenetic activation of the PVD may provide further evidence for the necessity of DEG/ENaCs in this structural plasticity, since neuron activation by ChR2 bypasses the activation of mechanically gated DEG/ENaC channels. *Figure 7* depicts our working model, where the amount of mechanosensory stimulation, in crowded or isolated conditions, affects the expression and localization of MEC-10 and DEGT-1 in different compartments of the PVD. Localization of MEC-10, probably by forming higher-order complexes with other DEG/ENaCs, can affect the structure of the PVD at the level of the dendritic tree. The morphological and behavioral phenotypes shown here were rescued by a PVD-specific expression of MEC-10, supporting such a cell autonomous mechanism. The experience-induced structural plasticity seems homeostatic at the individual branch dynamics level in the adult. In contrast, we show that the behavioral response to harsh touch is modulated by mechanosensory experience during development and by the presence of DEG/ENaCs, and perhaps involves other neurons downstream. Based on optogenetic stimulation, we presume that the behavioral plasticity is a pre- and/or postsynaptic property, mediated by DEG/ENaCs (*Hill and Ben-Shahar, 2018*) and related to neurotransmission modulation, independently of the structure of the PVD dendritic tree. These structural and behavioral plasticity are separated in time (adulthood vs. development) and space (dendrite vs. axon; *Figure 7*). We found no correlation or causation between the structure and the nociceptive function of the PVD. It is possible that such a link exists, but our physiological and behavioral outputs are not sensitive enough to detect it.

The PVD is a polymodal sensory neuron, and it has been shown that its dendrite structure is related to body posture, while the response to harsh touch is related to synaptic connection (*Tao et al., 2019*). Current assays for proprioception in *C. elegans* largely rely on video movement tracking, and it is possible the changes in the dendrite structure caused by sensory deprivation are too minor to cause any quantitative changes in body posture. Elegant work by *Tao et al., 2019* supports the role of MEC-10 in PVD-mediated proprioception, however has some discrepancies with our results with regard to harsh touch response, which may be related to the different methodologies used, as well as to the animals' age and mechanosensory experience (*Inberg et al., 2018*).

Calcium imaging reveals PVD responds to fast changes in mechanical stress, which includes pressure onset as well as offset (*Figure 6Aiii*, *Figure 6—figure supplement 2*, *Video 7*). We found PVD to be insensitive to slow stimuli and adapt under constant mechanical body deformation. This adaptation may facilitate the response to a wide dynamic range of stimuli applied to the body wall and is thus an essential feature of mechanical nociception. While TRNs respond to both the application and release of a step stimulus (*Eastwood et al., 2015*), other mechanosensory channels in different mechanosensitive systems respond preferentially to either on or off instead of a symmetric response to both (*Katta et al., 2015*). Interestingly, previous calcium imaging studies on PVD have not described the response to offset (*Chatzigeorgiou et al., 2010*; *Husson et al., 2012*; *Tao et al., 2019*; *Cho et al., 2017*; *Cho et al., 2018*). One possible explanation to our new observation may be due to the fact that we are the first to apply 'sawtooth' stimuli to the PVD with a slow increase such that the activity is not induced. An alternative explanation is that in previous work, the onset and offset were too close (e.g. for a 0.5-s step as in *Tao et al., 2019*), such that the onset and offset could not be differentiated.

Our results indicate that sensory experience does not alter the magnitude of the calcium activation (*Figure 6*), as tested using three different

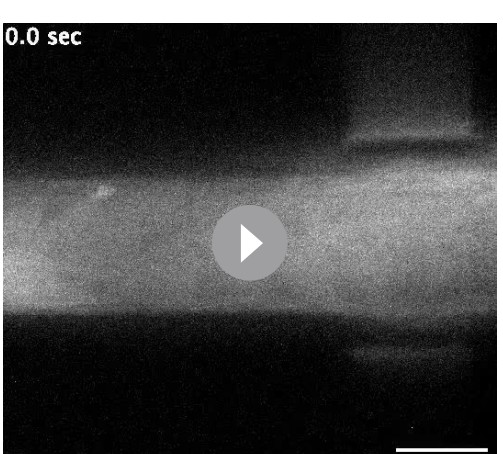

**Video 7.** Calcium dynamics in the PVD is sensitive to both onset and offset signals. PVD cell body received ipsilateral application of a mechanical stimulus and responded by increase in calcium signal for both the onset and offset stimulations. See *Figure 6—figure supplement 2* for more details. Scale bar, 40 μm.
https://elifesciences.org/articles/83973/figures#video7

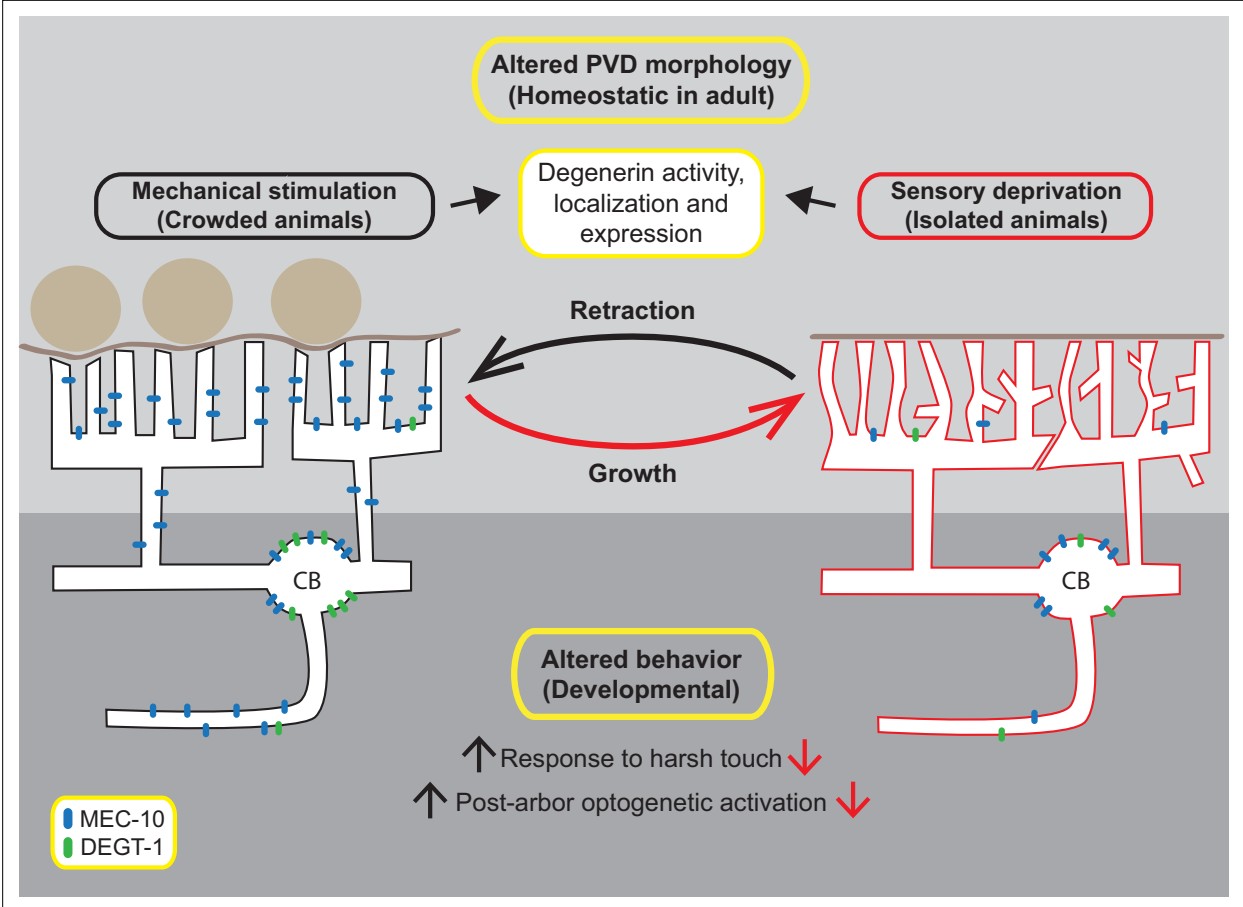

**Figure 7.** Model of experience-induced structural and behavioral plasticity. During adulthood, mechanosensory signals maintain the structure of the PVD in crowded animals, with straight quaternary branches and fewer ectopic branches. Sensory deprivation (in isolated animals) results in ectopic dendrites, wavy branches and defects in candelabra self-avoidance. During development, mechanosensory experience alters the response to harsh touch and the crawling gait of the worm (*Inberg et al., 2021*), possibly through changes in MEC-10 localization in the axon and mediated by other DEG/ENaCs. Mechanosensory stimuli are a driving force for changes in the compartment-specific localization of MEC-10 and DEGT-1 in the PVD, which may affect the structure of the PVD. MEC-10 is represented in blue, DEGT-1 is represented in green.

The online version of this article includes the following source data and figure supplement(s) for figure 7:

**Figure supplement 1—source data 1.** Original data file for *Figure 7—figure supplement 1* on side of plate contact alone over several hours is insufficient to elicit a difference in ectopic branching.

**Figure supplement 1.** Side of plate contact alone over several hours is insufficient to elicit a difference in ectopic branching.

stimulation profiles. These findings support additional effects downstream to the dendritic structural plasticity, which govern behavioral plasticity and add another layer of support to the separation in time and space between the structure of the PVD and its function.

Somatosensory activation in vertebrates plays a prominent role in shaping the structural and functional properties of dendritic spines, mainly studied in the central nervous system (*Györffy et al., 2018*; *Holtmaat and Svoboda, 2009*; *Xu et al., 2009*; *Yang et al., 2009*). In contrast to mammalian cortical neurons, much less is known about sensory neurons' degree of plasticity and the molecular mechanisms utilized during adulthood. Here we suggest that degenerins mediate mechanosensation-induced dendritic growth in sensory dendrites. The dendritic plasticity we described bears resemblance to the activity-dependent effect of glutamatergic signaling and NMDA receptors. Activity via NMDA affects dendritic spines as an upstream mechanism of cell signaling, resulting in structural modifications (*Nägerl et al., 2004*; *Star et al., 2002*; *Stein et al., 2021*; *Zhang et al., 2015*). It is possible that degenerins mediate mechanosensory signaling sensation, by activating cationic gradients (*Kellenberger and Schild, 2002*), leading to activation of downstream intracellular signaling pathways (*Ghiretti et al., 2014*; *Sin et al., 2002*; *Vaillant et al., 2002*; *Zhou et al., 2006*), which in

turn stabilize local, actin-mediated (*Halpain, 2018*; *Luo, 2002*; *Zuo et al., 2005*) structural plasticity in the PVD dendritic branches. In *Drosophila*, at the epithelium, mechanosensitive ion channels, together with E-cadherin–catenin complexes and calcium sensing mechanisms affect epithelial morphogenesis (*Roy Choudhury et al., 2021*). In parallel, DEG/ENaCs may also modulate pre- and postsynaptic homeostatic signaling in the harsh touch circuit, as has been shown in neuromuscular junctions (*Younger et al., 2013*). While this study focuses on degenerins activation in the PVD with emphasis on dendritic structures, future studies may establish the pre- and postsynaptic mechanisms which act downstream, on the transcriptional and translational levels. These directions for future studies have the potential to increase our understanding of the mechanisms that couple sensory experience to structural dendritic plasticity.

In summary, we propose that the combinatorial actions of DEG/ENaCs have mechano-signaling functions mediating plasticity in sensory dendritic trees, and provide mechanistic insights into dendritic structural responses to sensory experience in adulthood and the behavioral consequences of such adaptations during development.

## Materials and methods

### Strains

Nematode strains were maintained according to standard protocols (*Brenner, 1974*). The list of the strains is presented in *Supplementary file 1a*. Strains of the DEG/ENaCs family obtained from the CGC (JPS282: *asic-1*(*ok415*) I, ZB2551: *mec-10*(*tm1552*) X, and VC2633: *degt-1*(*ok3307*) V) were crossed with BP709: *hmnIs133[ser-2Prom3::kaede]*. The validation of F2 homozygotes for the DEG/ENaCs deletions (including single, double, and triple mutants) was performed by PCR amplification of the genomic area containing the deleted region.

### Primers for multiplex PCR

The list of the primers used is presented in *Supplementary file 1b*.

### Spinning disk confocal microscopy

Prior to imaging, the worms were mounted on a 10% agar pad placed on a microscope glass slide, in 1 µl polystyrene bead suspension (100 nm diameter; Polysciences, Inc) for their mechanical restraint, and sealed with a coverslip for complete physical immobilization (*Kim et al., 2013*). The PVD neuron was visualized using a Yokogawa CSU-X1 spinning disk, Nikon eclipse Ti inverted microscope, and iXon3 camera (ANDOR). Images were captured with MetaMorph, version 7.8.1.0. For each worm, a sequential z-series image stack (step size of 0.35 µm) was obtained with an oil Plan Fluor 40X (NA 1.3) lens around the PVD cell body, encompassing approximately 50–100 µm segments both anteriorly and posteriorly.

### Data analysis

The analysis of the PVD structure was performed for the area surrounding the cell body. All images were analyzed with ImageJ, version 1.48 (NIH), in TIFF format, by producing a maximal intensity z-series projection and converting it to negative form (invert lookup table) for improved visibility. Ectopic branching was defined as described previously (*Häusser and Mel, 2003*). Briefly, non-ectopic branches form the 'ideal' WT candelabrum of the late L4 stage, whereas excess branches, which create non-quaternary terminal ends, are considered ectopic, as illustrated with dashed lines for ectopic branching in *Figure 2A*. The total number of ectopic and non-ectopic branches was quantified for each image, and presented as a fraction (ectopic/non-ectopic branches).

The geometry of each quaternary branch was defined in the following manner: Straight geometry – all the pixels that constitute the branch are positioned on a straight line generated with ImageJ. The width of the line (1 pixel) was constant for the entire sets of experiments. The number of straight quaternary branches was divided by the total quaternary branches in the image for each worm and presented as a percentage. The analysis was done only for worms which did not move through the z-series. Moving worms were excluded from the experiment. Self-avoidance defects – the number of events where two adjacent candelabra overlapped (no gap formation) was divided by the total

number of gaps between the candelabra within the frame (*Figure 2A*, *Figure 4 – figure supplement 3*). The self-avoidance values are presented as percentage.

## Behavioral procedures

### Harsh touch assay

After 72 hr, adult worms (both isolated and those from the crowded plate as described below) were transferred using an eyelash each to a new agar plate, freshly seeded with 150 µl OP50 (about 16 hr after seeding). This step is required to avoid a thick edge of the bacterial lawn, which might interfere with harsh touch response measurement. After ~45 min in the plate, the non-moving worms were prodded with a platinum wire posterior to the vulva, above the interface between worm's body and the agar plate (*Way and Chalfie, 1988*), every 10 s, and the number of responses to harsh touch was counted. Animals with a functional PVD moved, sometimes backing up. More than one response constitutes a responsive animal. The non-responding worms were defined if two prodding events were observed sequentially without response. The percentage of responding worms was calculated for each genotype and treatment. The experimenter was blind to both the genotype and the treatment- crowded or isolated.

### Isolation of embryos

To establish the method for conspecifics-based mechanosensory stimulation, we performed several calibration experiments with different population densities (250 embryos, progeny of 15 adults, progeny of 30 adults) to test conditions for the crowded plate.

We found a gradual increase in the crowding effect and reduced variation in the measured parameter. Following that, we decided to work with the progeny of 30 crowded adults as a source of mechanosensory signal.

The worm isolation procedure was based on previous work (*Rose et al., 2005*) with a few modifications, as indicated. Isolated animals were grown on 6 cm agar plate with 150 µl of OP50 *E. coli*, while crowded plate worms were grown on 600–700 µl OP50 to prevent starvation. The embryos and adult worms were isolated with platinum wire. The plates were sealed with one layer of Parafilm M and placed into a plastic box, at 20°C, for the entire experiment.

Three experimental groups were used for the 72 hr (96 hr of experiment was performed only for *mec-4* worms, since they were L4 – very young adults at 72 hr) isolation experiment: (1) Single isolated embryos; (2) Crowded worms – the progeny of 30 young, non-starved, adults (approximately 7000–9000 worms in different developmental stages, without approaching starvation); (3) Crowded adult worms that were isolated for a certain amount of time as adults.

After 48/72/96 hr (according to the experiment), age-matched worms from each group were transferred to 10% agar pad slides for imaging, as described above (Spinning disk confocal microscopy) (*Rose et al., 2005*).

### Isolation of adults for 2, 5, 15, or 24 hr

Crowded plates were prepared as described above, with progeny from 30 adult hermaphrodites. Animals were separated into individual plates for the desired time window before 72 hr have passed (i.e. after ~70 hr for 2 hr isolation, ~67 hr for 5 hr isolation, etc.). Worms were isolated using an eyelash into new plates containing 150 µl of OP50. At 72 hr, the PVD of isolated animals was imaged and compared with age-matched animals which remained in the crowded plate of origin (see *Figure 1B*).

### Optogenetics stimulation for harsh touch

Crowded and isolated worms (ZX819: *lite-1(ce314) X; zxIs12[pF49H12.4::ChR2::mCherry pF49H12.4::GFP]*) were grown on OP50 with 100 µM All Trans Retinal (ATR, Sigma R2500), in order to obtain functional Channelrhodopsin (*Husson et al., 2012*) (as a control we tested the response of worms that were grown on ethanol alone, no response was detected for these worms). After 72 hr the worms were singly mounted with eyelash on a chunk (1 cm × 1 cm) of agar that was mounted on a microscope glass slide. The agar contained fresh but dry OP50, with 100 µM ATR. About 30 min following the transfer the worms were tested for the response to light. Worms were stimulated at 488 nm wavelength, with laser intensity of 40% and exposure time of 100ms, with 10X Plan Fluor

(NA 0.3) for ~1 s and the forward acceleration response was measured. The microscope system is as described above. The experimenter determined the presence or the absence of forward acceleration in response to light activation.

### Optogenetics stimulation of isolated worms
Isolated eggs were grown as described at the 'Isolation of embryos' section.

Isolated worms (ZX819: *lite-1(ce314) X; zxIs12[pF49H12.4::ChR2::mCherry pF49H12.4::GFP]*) were grown on 150 μM OP50 with 100 μM ATR (*Husson et al., 2012*) or 0.3% ethanol as control. After 72 hr both the crowded and isolated worms were singly mounted with eyelash onto a cube (with surface area of approximately 1 cm², depth of 0.5 cm) of agar layered with fresh, fully dried, OP50 with 100 μM ATR or 0.3% ethanol as a control group. The worms were stimulated with 488 nm wavelength, with laser intensity of 40% and exposure time of 100 ms, with 10X Plan Fluor (NA 0.3) for ~60 s, every 5 min during 4 hr. Stimulation and recording were performed with the microscope system described above.

Before each experiment, crowded ZX819 worms were tested for forward acceleration as a positive control to the functionality of the ChR2. After the end of 4 hr stimulation, the worms were mounted for imaging of the PVD at ×40 as described under 'Spinning disk confocal microscopy' section.

### Isolation with chemical stimulation
The *glp-4(bn2)* mutants which are sterile at 25°C were used (~40 worms for each plate), in order to prepare conditioned/chemically stimulated plates (*Maures et al., 2014*) prior to growing isolated animals. The *glp-4* mutants were transferred at early larval stage (L1 and L2) to a new agar plate for 96 hr at 25°C. After the removal of the *glp-4* mutants, the embryo isolation procedure was used, as described before at 20°C.

### Isolation with glass beads
Single embryos were isolated to agar plates with 150 μl OP50 and 2.5 g of glass beads (1 mm diameter) were placed on the OP50 lawn in the middle of the plate. The worms were isolated for 72 hr and tested for response to harsh touch as described above.

### Pharmacology
Amiloride hydrochloride hydrate (Sigma, #A7410) 1 M stock solution in DMSO was stored at −20°C. A final concentration of 3 mM amiloride in 0.03% DMSO was prepared in OP50 bacteria (LB medium) and seeded on NGM plates. 650 μl of the OP50 mixture was seeded on each plate. As a control, 0.03% DMSO was added to OP50 bacteria. For each plate (control 0.03% DMSO or 3 mM amiloride) 30 non-starved adult worms were added. After 72 hr at 20°C the progeny of the 30 adults were tested as young adults for their PVD morphology and their response to posterior harsh touch, as described in the previous sections.

### Anesthetics and long-term imaging
In addition to immobilization of worms with 100 nm polystyrene beads as described in the 'Spinning disk confocal microscopy' section, two additional methods for long-term imaging and pharmacological effects assays were used: 1% tricaine (Sigma, A5040) in M9 buffer or a mixture of 0.01% tetramisole (Sigma, T-1512) and 0.1% tricaine in M9 buffer were utilized.

Worms were exposed to the anesthetics (in a glass well) for ~20 min until paralyzed, then transferred to 3% agar pads with 1 μl of the anesthetics. Sequential *z*-axis image series (0.6 μm step size) were taken, as described above, every 5 min for 2–3 hr.

### Imaging individual worms – crowded to isolated
Worms were grown at crowded conditions, as described above. At ~24 hr of adulthood, individual worms were placed in a 1% tricaine solution in M9 buffer for 10 min until paralyzed and transferred to 3% agar slides with 1 μl 1% tricaine. Following a short PVD imaging session, the worms were recovered from the slide with M9 and isolated to a new plate for 4 hr. Each 4-hr-isolated animal was anesthetized once again with 1% tricaine for 10 min prior to PVD imaging.

## Analysis of DEG/ENaCs localization in the PVD

Two DEG/ENaCs protein constructs, *pF49H12.4::MEC-10*::mCherry and *pF49H12.4::DEGT-1*::mCherry (plasmids kindly provided by W. Schafer's lab, *Chatzigeorgiou et al., 2010*) were analyzed for their localization in the PVD, by comparing worms expressing the co-injection marker *punc-122*::GFP raised in crowded or isolated conditions, in a similar behavioral assay as described in *Figure 1B*. A z-series (step size of 0.35 µm, immobilization performed with polystyrene beads, as described in the microscopy section) of the area around the cell body of the PVD was obtained with a 60X Apochromat (NA 1.4) lens. The images (after maximal intensity projection of the z-series) were encoded so the analysis was performed in a blind manner. The presence or absence of fluorescent signal was examined in three compartments: the cell body, the quaternary branches and the axon of the PVD.

## Rescue strains

Worms from BP1022 (*mec-10(tm1552) X; hmnIs133[ser-2Prom3::Kaede]; him-5(e1490) V*) were injected into the gonad with a rescuing plasmid for *mec-10*, with a PVD-specific promoter (pWRS825: *ser-2Prom3::mec-10* genomic) kindly provided by W. Schafer's lab (*Chatzigeorgiou et al., 2010*). The injection mix contained *myo-2p::GFP* (20 ng/µl) as a co-transformation marker and pWRS825 (80 ng/µl). For both behavioral and structural characterization, the *him-5; mec-10* strain, with and without rescuing plasmid, shared the same crowded plate and were differentiated by the presence or absence of the co-injection marker.

## Effect of plate on PVD in contact with agar and the opposing side

Embryos were singled into plates with 150 µl OP50 bacteria at 20°C, and analyzed as young adults 72 hr later. Every animal was transferred using an eyelash into freshly seeded plates (<24 hr), and noted for its side orientation. Once on these experiment plates, animals were observed for their side orientation at 30-min intervals for a minimum of 3 hr prior to microscopy.

For microscopy of both PVD neurites, while few animals were removed from the slide and repositioned using an eyelash, most animals were imaged using a coverslip 'sandwich' method (adapted from *Sulston et al., 1983*), whereby the animal is pressed on an agar pad between two large coverslips, which are subsequently flipped as a single unit and observed from the other side. Briefly, a thin 3% agar pad was pressed onto a 24 × 50 mm coverslip (170 ± 5 µm thick), placed on a standard microscope slide for ease of handling. Animals were singly placed in a drop of 0.05% tetramisole (Sigma, T-1512) in M9 on the agar pad using an eyelash, and a second, similar coverslip was immediately placed above. The two coverslips were sealed shut onto the supporting slide using two thin strips of marking tape, and once the animal was immobilized the PVD on the side closest to the objective was imaged as described above (see Spinning disk confocal microscopy). For viewing the opposite neurite, the tape was carefully peeled and the two coverslips flipped as one, taped again, and similarly imaged. Note that this method inevitably makes the second side imaged to be fainter due to the laser penetrating through the agar pad layer itself; laser intensities were typically increased by up to 5% in order to compensate for the decrease in fluorescence.

## Micromechanical experiments and calcium image recordings

Micromechanical experiments were performed in microfluidic devices which have been prepared as described using a 15:1 Polydimethylsiloxane (PDMS) base-polymer/curing agent ratio (*Fehlauer et al., 2018*; *Setty et al., 2022*), by replicating a design presented in *Nekimken et al., 2017*. Prior to the experiments, single animals raised in isolated conditions or multiple animals raised in the crowded conditions were loaded into the device as described (*Fehlauer et al., 2018*). After gently pushing individual animals into the trapping channel, a sequence of three different stimulus profiles (2 s each, 3 bar max pressure, immediate 'step', sawtooth [or gradual 'ramp'] or 10 Hz vibration 'buzz') were applied (*Nekimken et al., 2017*) each separated by 20 s. The pressure–deformation relation was previously characterized (*Setty et al., 2022*). Accordingly, we found a deformation of 15 µm with a 3-bar pressure on the channel. In all cases, the animals ended up with the lateral sides facing the side walls of the chip, providing a ventral or dorsal view of the body. Stimuli location was within 200 µm of the PVD cell body (either anterior or posterior), and ipsilateral or contralateral.

Imaging was performed using a ×40 (NA1.1) water immersion lens on an inverted Leica DMi8 equipped with a Hamamatsu Orca Flash 4 camera, and a fluorescence cube with a multiband dichroic mirror (Semrock Quadband FF409/493/573/652). Calcium-sensitive and -insensitive channel were divided into its spectral components using a Hamamatsu Gemini W-View beamsplitter with a 538-nm edge dichroic (Semrock, FF528-FDi1-25-36) and projected on either half of the camera chip. A Lumencor Spectra X LED light source (using the cyan-488 and green-555 LED) was used to excite the Ca-sensitive and -insensitive dye, respectively, and slaved to the camera acquisition with an exposure time of 100 ms in HCImage Software (version 4.4.2.7). Likewise, the pressure application (ElveFlow OB1 with an 8-bar channel) was synchronized to the camera acquisition using the SMA trigger out, with a delay equal to the pre-stimulus period.

The whole procedure including loading, focusing, and stimulation did not take longer than 5–10 min per animals to avoid pre-stimulus adaptation. All animals remained viable after the procedure, as visualized by vivid and lively thrashing in the buffer.

## Analysis of calcium traces after stimulation

All quantifications were performed on the PVD cell body after ipsilateral application of the mechanical stimulus. No dependence of the calcium signals on the anterior/posterior position was noticed. Contralaterally applied stimuli lead to a strongly reduced response, without detectable difference between the two conditions (isolated, crowded), but were not further analyzed. To extract calcium signals from the cell body, a procedure described in *Porta-de-la-Riva et al., 2023* was followed. Calcium traces were background subtracted, baseline corrected, and normalized by the first 10 s preceding the first mechanical stimulus. Results are displayed as fold change, compared to baseline calcium levels.

## PVD basal activity

To image spontaneous calcium traces in PVD, we used worms that were raised in isolated or crowded conditions, as described above. Using a drop of S-basal buffer, adult worms were inserted in a dual olfactory chip (*Gat et al., 2023*; *Karimi et al., 2024*), with a flow rate of 0.005 ml/min. Imaging was done with a Zeiss LSM 880 confocal microscope using a ×40 magnification water objective. When the worm was properly located inside the chip with minimal movement, PVD was imaged for 1:30 min, with an imaging rate of 6.667 Hz.

For analysis, the GCaMP6s fluorescence intensity was measured using FIJI. ROIs of PVD soma were picked manually to extract mean intensity values.

Data analysis was performed using Python 3.10. For each worm, the baseline fluorescent level ($F0$) was calculated by averaging the mean values of 100 frames in the beginning of each recording. Then, for each frame, $\Delta F$ was calculated by subtracting $F0$ from the value of that time point, and the result was divided by $F0$, to normalize the differences in the fluorescence baseline levels between individuals ($\Delta F/F0$). For *Figure 6—figure supplement 1C*, the derivative of $\Delta F/F0$ was calculated and averaged across worms in each time point. Then, a histogram representing the distribution of the average derivative was plotted.

## Statistics and data plotting

At least two independent experiments constitute the dataset described in each figure.

For the morphological characterization of the PVD, the results are expressed as means (blue circle) ± SEM. In the boxplot (first and third quartiles) the upper whisker extends from the hinge to the highest value that is within 1.5 * IQR (inter-quartile range), the distance between the first and third quartiles. The lower whisker extends from the hinge to the lowest value within 1.5 * IQR of the hinge.

The statistical analyses were performed with SPSS software (IBM, version 20) and 'R package'. Two-tailed tests were performed for the entire data sets.

Since for many experiments the distribution of the data was not normal, nonparametric tests were used: Mann–Whitney test for comparison between two independent groups. Kruskal–Wallis test was used for multiple comparisons for more than two groups.

For proportions (percentage worms) ± standard error of proportion was calculated.

Fisher's exact test was used for analysis of differences in proportions. To estimate the variability in proportion we calculated the standard error of proportion: $\sqrt{\frac{(1-p)\cdot p}{n}}$.

The dot plot figures were prepared with 'R package', the bar charts with Microsoft Excel software. Final figures were prepared with Adobe Illustrator CS version 11.0.0.

## Discriminant analysis

Eight different strains (WT and seven DEG/ENaCs), with two treatments (crowded and isolated worms) for each strain were analyzed for Linear discriminant analysis for morphological characteristics, to evaluate similarity between different strains and treatments. Each worm in the dataset was characterized by the three morphological characteristics (the fraction of ectopic branching, the percentage of straight quaternary branches, and the percentage of self-avoidance defects). The centroid for morphological characterization was calculated for each condition and represented by a square. Data from independent harsh touch experiments are shown for each group. The analysis was performed using SPSS 20.

## Acknowledgements

We thank current and former lab members for their intellectual and technical support. Veronika Kravtsov, Sagi Levy, Anna Meledin, Tom Shemesh, Shay Stern, Yehuda Salzberg, and Alon Zaslaver for critically reading and commenting on the manuscript. Ehud Ahissar, Dan Cassel, Michel Labouesse, and Kang Shen for fruitful discussions. William Schafer, Max Heiman, Hannes Bülow, Yehuda Salzberg, and Alexander Gottschalk for plasmids and strains. Some strains were provided by the *Caenorhabditis* Genetics Center (CGC), which is funded by NIH Office of Research Infrastructure Programs (P40 OD010440). This work was supported by grants from the Israel Science Foundation (442/12 and 257/17, BP), Adelis Fund (2023479, BP), and the Ministry of Science and Technology (3-13022, BP). MK acknowledges PID2021-123812OB-I00 and the CNS2022-135906 project funded by MCIN/AEI/10.13039/501100011033/FEDER, UE, and the HFSPO through the RGP021/2023.

## Additional information

### Funding

| Funder | Grant reference number | Author |
| --- | --- | --- |
| Israel Science Foundation | 442/12 | Benjamin Podbilewicz |
| Israel Science Foundation | 257/17 | Benjamin Podbilewicz |
| Adelis Fund | 2023479 | Benjamin Podbilewicz |
| Ministry of Science and Technology, Israel | 3-13022 | Benjamin Podbilewicz |
| MCIN/AEI/10.13039/501100011033/FEDER, UE | PID2021-123812OB-I00 | Michael Krieg |
| MCIN/AEI/10.13039/501100011033/FEDER, UE | CNS2022-135906 | Michael Krieg |
| Human Frontier Science Program | RGP021/2023 | Michael Krieg |

The funders had no role in study design, data collection and interpretation, or the decision to submit the work for publication.

### Author contributions

Sharon Inberg, Conceptualization, Resources, Data curation, Formal analysis, Validation, Investigation, Visualization, Methodology, Writing – original draft, Writing – review and editing; Yael Iosilevskii, Conceptualization, Data curation, Formal analysis, Validation, Investigation, Visualization,

Methodology, Writing – review and editing; Alba Calatayud-Sanchez, Data curation, Formal analysis, Investigation, Methodology, Writing – review and editing, Designed, performed and analyzed the calcium dynamics in response to harsh touch; Hagar Setty, Data curation, Formal analysis, Investigation, Methodology, Writing – review and editing, Designed, performed and analyzed the calcium baseline experiments; Meital Oren-Suissa, Data curation, Formal analysis, Supervision, Funding acquisition, Investigation, Methodology, Writing – review and editing, Designed, performed and analyzed the calcium baseline experiments; Michael Krieg, Data curation, Formal analysis, Supervision, Funding acquisition, Investigation, Methodology, Writing – review and editing, Designed, performed and analyzed the calcium dynamics in response to harsh touch; Benjamin Podbilewicz, Conceptualization, Formal analysis, Supervision, Funding acquisition, Writing – original draft, Project administration, Writing – review and editing

**Author ORCIDs**
Sharon Inberg ⓘ http://orcid.org/0000-0003-1405-2779
Yael Iosilevskii ⓘ http://orcid.org/0009-0004-1156-5581
Benjamin Podbilewicz ⓘ https://orcid.org/0000-0002-0411-4182

**Decision letter and Author response**
Decision letter https://doi.org/10.7554/eLife.83973.sa1
Author response https://doi.org/10.7554/eLife.83973.sa2

## Additional files

**Supplementary files**
Supplementary file 1. Lists of strains, transgenes and primers used in this work. (**a**) List of strains and transgenes used in this work. (**b**) List of primers used in this work.

MDAR checklist

**Data availability**
All data generated or analyzed during this study are included in the manuscript and supporting file. Strains, plasmids, and other reagents are available upon request.

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
