## [Editor Report]

This is an important study on neuronal plasticity demonstrating that mechanosensory isolation of *C. elegans* nematodes induces homeostatic structural changes in the dendritic tree and differential response to mechanical stimulation. Convincing evidence is provided to show that both structural and behavioral outcomes are mediated by degenerins – a highly conserved family of ion channels. The study will be of broad interest to the neuroscience community.

---

## [Decision Letter]

[Editors' note: this paper was reviewed by Review Commons.]

Based on the previous reviews and the revisions, the manuscript has been improved but there are some remaining issues that need to be addressed, as outlined below:

1. There is consensus among the reviewers that Calcium imaging is needed to support the conclusions of this work.

2. The authors are also encouraged to examine pre- and post-synaptic markers in PVD upon sensory deprivation.

3. In your response letter, please address the reviewer comments listed below.

Reviewer comments

Reviewer 1: I remain uneasy about the adult contributions to the phenotypes seen, and the fact that siblings don't seem to have the effect- it makes me wonder what it is that is actually being studied. I also agree that calcium imaging could be done through collaboration, and would add quite a bit. The main question in my mind is whether they have demonstrated that the effects are due to mechanosensory stimulation.

Reviewer 2: The authors have responded to all the queries made by the reviewers. It is understandable that sensory experience-dependent behavioral plasticity and dendrite arborization is not related. That is also suggested by another manuscript (Tao et al. 2019 Dev Cell) that dendrite structure is related to body posture, whereas harsh touch is mediated by the synaptic connection. PVD acts as a polymodal sensory neuron.

The authors need to discuss the point clearly that the changes in the dendrite structure caused by the sensory deprivation might be very small to cause any quantitative changes in the body posture. As there is not a great assay developed for proprioception in worm.

However, I noticed that in the existing contexts discussed in the paper, the authors could address the experimental suggestion made by the referees. Especially, pre or post-synaptic changes in PVD neurons upon sensory deprivation. or Calcium dynamics in PVD or/and in the neuron postsynaptic to PVD.

This perhaps could improve the manuscript.

Reviewer 3: The authors have satisfactorily addressed many of the reviewer's comments. In the response to reviewers, the authors defended the lack of a clear mechanism by indicating that this manuscript focused on describing the behavioral and morphological phenomena of isolation and showing the involvement of two genes; however, because the behavioral and the anatomical changes do not correlate, this negatively impacts the overall quality of this manuscript. It could be particularly informative to examine changes in pre- and post-synaptic markers and/or the calcium dynamics in the PVD neurons, given that their optogenetic data suggest the behavioral effects of isolation appear to be post-sensory, and these experiments could potentially answer the question of whether the behavioral changes are mediated by changes in cell excitability or synaptic strength. There are a fair number of labs that have performed calcium imaging experiments on PVD who might be willing to collaborate- this could be a potential avenue of the investigation of the mechanisms. Because the 2 measures do not correlate the paper is not as impactful as it would be if there was some understanding of why they did not.

A new issue is that in the response to reviewer #1, the data shown in Figure 1 is problematic. The reviewer asked whether there were behavioral differences in the populations of worms reared in different conditions, however, Figure 1 described the dendritic structural differences, and no behavioral data was presented. Additionally, data in Figure 1 appear to suggest that some effects of crowding are dependent on the parents (1AandB), while others are dependent on the siblings (1C) – it is important to see the behavioral effects in these same populations and some discussion of these observations.

---

## [Author Response]

Reviewer #1 (Evidence, reproducibility and clarity (Required)):This manuscript describes the consequences of placing *C. elegans* nematodes in isolation or in a dense population. The authors show that crowding conditions influence the behavioral response of animals to harsh prodding with a platinum wire, the morphology of the PVD neuron thought to mediate this response, and the expression/localization of ion channels within PVD. The authors show that the behavior and structure under various conditions do not correlate with each other. The authors identify DEG-family channels required for the differences in behavior and PVD morphology in various settings.While the nervous system must remain properly wired to reproducibly respond to stimuli, plasticity to stimuli is also important, to allow survival in changing environments. The authors here investigate several plastic aspects of the nervous system that govern mechanosensation, and identify relevant molecular players. The studies are generally well done, and the interpretations are appropriate. The paper does not provide mechanistic insight into how changes in behavior, neuron shape, or channel expression are governed, and is therefore largely phenomenological in nature. Nonetheless, the plasticity recorded seems to me important to describe, as it sets up a new setting for investigating such phenomena.1. In Figure 1B, it appears that on the crowded plate there are adults as well as embryos.It is possible that adult presence somehow affects touch responses of developing larvae. The appropriate control to the single embryo/plate would be plates with multiple embryos but no adults.

Indeed, crowded plates contain adult worms crawling in the plate together with larvae. In preliminary experiments, that were part of our calibrations (shown below in Figure 1), we have analyzed several population densities to determine the minimal number of animals necessary to establish the “crowdedlike” state. Following the reviewer’s comment, we decided to briefly describe the calibration process of different population densities in the manuscript (described in the manuscript under Methods: Isolation of embryos; lines 542-548)“. To establish the method for conspecifics-based mechanosensory stimulation, we performed several calibration experiments with different population densities (250 embryos, progeny of 15 adults, progeny of 30 adults) to test the conditions for a crowded plate. We found a gradual increase in the crowding effect and reduced variation in the measured parameter. Following that, we decided to work with the group of 30 crowded adults as a source of mechanosensory signal.

Author response image 1 shows that 250 grouped eggs behave similarly to a single isolated egg, except for self-avoidance defects. Obtaining a clean suspension of embryos at this amount usually involves a hypochlorite treatment, which we prefer to avoid due to potentially toxic effects. Since we have evidence that pheromones are not involved in crowding-influenced touch response, we opted to keep the protocol including adult presence, as this should serve to enhance the cumulative mechanosensory signals on the plate. The crowded plates contain a wide range of worms at different ages. The entire heterogeneous population serves as a source for mechanosensory signals.

**Author response image 1. sa2fig1:** PVD morphology is dependent on the density of the worms in the plate. Worms were grown at different population densities, either as progeny from 30 adults on the plate, progeny of 15 adults or 250 eggs collected from the source plate (crowded 30 adults, n=28; crowded 15 adults, n=12; 250 eggs, n=19; isolated single worm, n=26) for 72 h. PVD morphology was quantified at ~24 h of adulthood. (A) Animals raised as isolated or from 250 eggs in the plate show an increased amount of ectopic branching compared to crowded 30 worms. (B). Straight quaternary branches were reduced in all other plate densities assayed when compared with the progeny of 30 adults. (C) Isolated worms show increased selfavoidance defects compared with crowded progeny of 30 worms. Mann Whitney test: *p<0.05, **p<0.005, ***p<0.0005.

2. For the harsh touch assay, it seems that the force applied to each animal was not measured. If the forces applied were skewed in some way between experimental and control, this could be a confounding factor. To avoid this, mechanosensory stimuli with predetermined forces (e.g. caliberated von Frey hairs) probably should be used.

Indeed, the force applied to the animal was not measured, but the assay was carefully calibrated in order to use the minimum strength which elicits a response, without harming the worms. We observed that the behavior of the worms remains normal (no freezing, slowing or problem with egg laying), and also the structure of the PVD was not affected. To eliminate the possibility of an operator error, an animal which did not respond was contacted a second time after ten seconds to confirm non-responsiveness. In addition, in order to avoid confounding factors and any unwanted biases related to the identity of the worms (crowded or isolated), the entire set of experiments for the harsh touch procedure was performed blindly. At least two independent experiments were performed for each genotype, using freshly prepared uniform assay plates (this is explained in the Behavioral procedures section on Harsh touch assay; Lines 530-541).

Reviewer #2The manuscript by 'Sharon Inberg' et al. tests the hypothesis that how sensory experience affects dendrite morphology in a neuron and how that might be related to functional output of the neuron. They use a combination of genetics, optogenetic, behavioral, and pharmacological approaches using PVD neuron of *C. elegans*, which has stereotypic dendritic branches. This neuron has different functional output such as sensation of harsh touch and proprioception. Their experimental outcome suggest an activity-dependent homeostatic mechanism for dendritic structural plasticity, that in parallel controls escape response to noxious mechanosensory stimuliThis is an interesting study and many interesting experimental attempts have been made to understand how changes in dendritic tree is related to the functional change with respect to harsh touch sensation. However, I have few suggestions to improve the mechanistic understanding dealt in this article.Major comments:1) One main conclusion is that sensory experience during development is important for harsh touch sensation during adulthood. In this paradigm, they are isolating the animal from time of hatching and testing behavior in 72h after hatching. So isolation time is 72h.However, similar isolation experiment in adulthood was done only for 24h duration. Therefore, I suggest that the authors consider increasing the time of isolation in adulthood and see whether this influence the harsh touch behaviors.

In our characterization of the minimal time frame necessary for inducing isolation-dependent alterations, we took particular care to obtain age-matched animals at the end of each protocol. This is crucial as aging has been shown to greatly influence the PVD dendritic structure (Kravtsov et al., 2017). This prerequisite meant we could not extend adult isolation time beyond 24 h, since our main assay at 72 h from egg means animals undergo larval development for ~48 h and an additional ~24 hours as adults, at the conditions that were used for the entire set of the experiments. Longer duration (>72h from egg) proved difficult to reliably maintain at the same crowding level without starving the plate. Moreover, it might be harder to recognize the age of the worms and to compare them reliably with isolated counterparts.

To reveal the specific contribution of the adult stage to the behavioral and physiological outcomes, we decided to grow worms at crowded conditions for 48 h (the entire larval stages) and then isolate them as very young adults for 24 h. In both isolated and adult-isolated procedures, the worms spent the first 24 hours of adulthood isolated, which enables us to compare between the two treatments. While harsh touch response was not affected, we have shown that isolation during the adult stage is sufficient to induce structural changes to the PVD.

2) The data is interesting that the decrease in harsh touch response due to developmental isolation does not corelate with the changes in dendrite branching. This is reinforced by the optogenetic stimulation experiment. It will be nice to have more evidence on the pre and post-synaptic changes caused due to sensory isolation. The authors might consider looking at the presynaptic components and post-synaptic markers.

Thank you for the important comment. The pre- and post-synaptic mechanisms which may contribute to the experience-dependent behavioral response to harsh touch are indeed very interesting, however the main focus at this work is the structural plasticity at the level of the dendritic tree. This aspect of neuronal structural plasticity is less studied, in contrast to the volume of work on synaptic plasticity. In addition, we asked whether dendritic structural changes are reflected in behavioral changes, and found no evidence for correlation between the two. As such, while this is a highly important avenue for future research, we believe it is beyond the scope of this paper.

3) Can authors discuss how the sensory experience, which is mediated by the activity of MEC-10 would lead to synaptic changes? Would it be transcriptional? They might consider looking at some activity dependent transcription factor such as CREB/FOS/JUN in this process. It would also be nice to have some discussion based on previous literature.

In the discussion, we elaborated on the possible downstream consequences of degenerin activation on both the dendrites and the synapses. Indeed, focusing on immediate early genes/transcription factors/ such as CREB/FOS/JUN and calcium sensing mechanisms together with their downstream targets, can be an interesting direction for a follow-up project.

Here we focused on the contribution of membrane bound degenerins, that are the upstream sensory components of the mechanosensory signaling pathway.

In the discussion, we added a comment about possible future directions of revealing the downstream mechanisms of degenerin activation (Lines 479-482). In addition, following the reviewer’s comment we elaborated more on mechanosensory signaling and the possible pathways which may be involved (Lines: 468-477).

4) The observation that changes in dendritic tree does not correlate with the nociception behavior is interesting. They have demonstrated that one can cause changes in the quaternary branches by various manipulation. However, these changes might be related to the other behavior involving proprioception. They have provided some data on crawling amplitude related to the developmental isolation in Figure 1 (related Supplementary). Can they evaluate these parameters in conditions when dendritic structure is changed? It is relevant in the context of the previous finding that dendritic release of neuropeptides were related to proprioception (Tao et al. 2019 Dev Cell).

We have attempted to correlate the branching morphology of the PVD with crawling parameters, by analyzing the movement of isolated worms and then imaging and blindly quantifying the PVD structure. While we did not examine quaternary geometry in these experiments, ectopic dendritic branching densities of orders 2, 3 and 4 failed to correlate with averaged movement parameters such as amplitude. Such a correlation, if present, is likely to be extremely noisy to detect due to variability in both structure and movement. At the moment, our preliminary results fail to indicate any correlation between the various branching orders and crawling parameters on the single-animal level, albeit the close association between the quaternary branches of the PVD with the body wall muscles. It is possible that deeper analysis of the structural, behavioral and synaptic changes will reveal such a correlation.

Minor comments:I suggest to remove statements that refers to "data not shown". In couple of occasions I noticed that.

We followed reviewer’s suggestion and removed this statement in line 431 and instead added a new to Figure S12.

Reviewer #2 (Significance (Required)):There are many evidences in vertebrate system that activity during early development is critical for proper wiring and functional output. Specially, using the visual system by Carla Shatz and other groups have demonstrated early sensory deprivation causes many changes in the neural circuitry.(L. C. Katz, C. J. Shatz et al. 1996 Scinece, C. S. Goodman, C. J. Shatz et al. 1993 Cell, Y. Zuo, et al. 2005 Nature). However, the evidence at the level of dendritic structural changes are lacking. The molecular details of this process is not completely clear till date.Therefore, this study draws a parallel using simple model *C. elegans*. The paradigm developed in this study might be very useful in deciphering the molecular mechanism in this process.Referees cross-commentingI agree with both the reviewers1 and 3 that there are points need to be cleared. This will improve the manuscript substantially.The reviewer# 3 correctly pointed out that how sensory experience modifies harsh-touch behavior at the cellular and circuitry level is not clear. This is similar to my suggestion to look at the pre and post-synaptic changes as well as looking at transcriptional regulation downstream to mec-10.However, the ca^2+^/GCaMP imaging might be a new direction, which involves having microfluidics technology in lab. Therefore, it can be optional depending on the availability of such resources in the author's lab.

We agree that experience dependent modifications and their effect on the circuitry level is an interesting direction. Our motivation here is to demonstrate that the structure of the PVD is plastic and can undergo structural modifications in response to external mechanosensory signals. We showed that experience dependent modifications affect the structure of the PVD via cell autonomous activity of MEC-10.

Pre- and post- synaptic mechanisms, as well as downstream molecular signals, are important avenues of future research, but go beyond the scope of this study, which focused on experience and degenerin-mediated plasticity at the dendritic level.

Indeed, ca^2+^/GCaMP imaging can open a new direction. Our pharmacological experiments with tetramisole and tricaine suggest that some intrinsic properties of the PVD are altered after isolation, compared with crowded worms. Unfortunately, our lab is not equipped for performing these experiments now.

Reviewer #3Summary:In the manuscript the authors aim to investigate the how sensory experience can impact dendritic structure and function. Using *C. elegans* and an isolation protocol and harsh-touch behavioral paradigm, the authors assess how sensory deprivation alters the animals response to harsh touch. By conducting a screen of amiloride-sensitive epithelial sodium channel mutants, the authors identify two channels (MEC-10 and DEGT-1) that mediate the isolation dependent changes in behavior. The authors assess how isolation impacts the structure of PVD dendrites, and whether isolation induced changes can be rescued with increased sensory experience. The authors find that isolation-induced structural abnormalities can occur within hours, and structural abnormalities can be partially rescued with increases sensory experience (if glass beads are placed on the plate). The authors attempt to identify a relationship between PVD structure and behavior but do not find a correlation. The authors then determine sensory dependant changes in the localization of MEC-10 and DEGT-1 in different areas of the PVD neuron. Lastly, the authors use optogenetics to investigate whether the isolation-induced changes in PVD dendritic morphology and degenerins localization are responsible for the observed behavioral plasticity and found that the mechanism for the behavioral plasticity is occurring downstream of sensory detection.Major comments:-Are the key conclusions convincing?•The authors provide some convincing evidence to support their conclusions, however additional experiments are needed to provide adequate support of their model and resolve unanswered questions scattered throughout the manuscript. The authors often jump from asking one question about the structure/function of PVD to an faintly related experiment without exploring the unexplained trends across mutants or environmental conditions (see comments for examples).-Should the authors qualify some of their claims as preliminary or speculative, or remove them altogether?•There is not sufficient evidence to support the authors model as there was no actual evidence showing the mechanisms driving the isolation-dependant changes in PVD structure or function. Further, the authors did not find a relationship between the structure of PVD and behavioral outcome. The authors may want to re-consider including their model within the manuscript or pursue further experiments to provide stronger evidence.

In this work we asked if and how mechanosensory experience can induce structural plasticity in the PVD neuron, and whether any structural changes are in turn correlated with altered behavior. Indeed, we failed to detect such a relationship. A correlation does exist between the crowded state and altered MEC-10 and DEGT-1 localization, with the more mechanistic evidence where PVD-specific MEC-10 expression rescues the *mec-10* mutant isolated-looking morphological phenotype of the crowded state. Simultaneously, isolated animals show several distinct behavioral alterations and respond differently to optogenetic PVD activation.

Indeed, we were unable to detect any correlation between the structural measurements that we performed and the various behavioral outputs (Figure 4A), nor establish a genetic pathway based on multiple degenerin mutant combinations (Figure S8, S9A-C). It is possible that such simple relationships do not exist or that our measurements are not sensitive enough.

As we explained throughout the manuscript, following the initial candidate gene approach screen, our main focus was on MEC-10 as a model gene to study the relationships between external mechanosensory signals to morphological modifications. We have shown that MEC-10 is both necessary and sufficient within the PVD to mediate experience -induced morphological and behavioral plasticity.

The model depicted in the paper was not intended to provide a comprehensive mechanism, but rather attempts to summarize the genetic information we have regarding mechanosensory induced alterations, at the level of PVD dendritic structure and degenerin protein localization. We chose to focus on the cell autonomous activity of MEC-10, together with animal behavioral output, to demonstrate that all aspects are affected, however neuron structure and function are not directly tied.

-Would additional experiments be essential to support the claims of the paper? Request additional experiments only where necessary for the paper as it is, and do not ask authors to open new lines of experimentation.•(241) The authors investigate whether PVD branching abnormalities can be rescued by place glass beads on the plate and found that it rescued branch straightness but did not rescue ectopic branching. Does placing isolated worms in a crowded environment fully rescue PVD branching abnormalities. Does environment richness affect the degree of rescue? (422) Similarly, in the discussion of their model the authors state that the amount of mechanosensory stimulation affects expression and localization of MEC-10 and DEGT-1 (crowded versus isolation). Did they authors investigate the effects of extreme crowding (increasing the number of worms on a plate above crowded plate levels) or partial crowding (decreasing the number of worms on a plate to be in-between crowded and isolated levels) on PVD structure or protein localization? This data would provide more evidence towards their model that the amount of mechanosensory stimulation drives this change.

Regarding the rescue experiment suggested by the reviewer, we haven’t tried to reverse the isolation-induced phenotype.

We agree that attempting to enrich the environment of an isolated animal by placing it in a crowded environment is an interesting experiment which we did not look into. We have performed preliminary assays using variable plate densities (Figure 1 in this response). The final methodology, utilizing 30 adult worms per plate, was the highest density we could reach for the experiment duration prior to plate starvation. Our preliminary results (Figure 1) show that reducing plate density leads to an isolated-like state, presumably due to insufficient or reduced collisions between conspecifics. In order to get a more stable mechanosensory environment, with higher probability to induce collisions, we decided to focus on the “30 crowded adults” group. Decreasing the density will reduce the crowding effect, while increasing the density of the worms may induce starvation, which we prefer to avoid.

In the methods, under “isolation of embryos” section, we added a brief description of our preliminary calibration and we explained the rational behind our decision (Lines 543-548).

•(284) The authors use pharmacological intervention (tetramisole and tricaine) to investigate how changing global neuronal activity affects the dendritic branching in PVD, however the effects of these drugs on behavior is not stated. It would be more convincing, and potentially provide more insight into the underlying mechanisms if the authors tested how genetically altering neuron activity affects PVD structure. Does knocking out receptors present on PVD such as osm-9 or glr-4 lead to differences in PVD branching or do branching abnormalities only occur when global nervous system activity is impaired?

Indeed, we used imaging to test the in-vivo effects of pharmacological intervention with tetramisole and tricaine on the structure of the PVD. From our experience, recovering the worms from the above-mentioned anesthetics will mask any effect on harsh touch response, considering the fact the drugs are eliminated from the body. Moreover, recovery of the worms will indicate elimination of the anesthetics from the body, which may interfere with any pharmacological conclusion.

We focused on the degenerin family of receptors as candidate genes that mediate experience dependent plasticity. Our rescue experiments with PVD-specific expression of MEC-10 indicate that specific alterations to an activity-inducing channel are sufficient to affect PVD structure. Finding additional receptors which contribute to the coupling between mechanosensory experience and dendritic morphogenesis is an interesting direction, that can be explored in future research.

•(358) The authors propose that the differential localization of mec-10 and degt-1 following isolation is driving the structural changes in PVD branches and potentially the function of PVD. It would be more convincing if the authors show that there are isolation induced changes in PVD activity using calcium imaging or observing changes in PVD output. The authors showed that optogenetic activation of PVD (therefore bypassing deg channel activation) still produces a decrease in harsh touch response in isolated animals compared to crowded animals. This finding does not provide support for the authors' hypothesis as it does not test whether the differential localization of the deg channels is important. If anything, the results show that the localization of mec-10 and degt-1 does not play a significant role in isolation-induced decrease in harsh touch response.

We agree with the reviewer that calcium imaging to see effects on PVD activity post-isolation could be a good approach. Unfortunately, our lab is currently not equipped for conducting calcium imaging experiments. Our optogenetic results may indeed indicate additional downstream mechanisms, which are beyond the scope of this paper. This is now mentioned in the discussion (lines 449-452).

•(362)The optogenetic experiments show that the isolation-induced decrease in harsh touch response is not dependant on the activity of deg channels, and that the isolation dependant changes in PVD branching are not solely dependent on PVD activity. The authors hypothesize that the mechanism responsible for the behavioral plasticity is occurring downstream of sensation. However, there is no exploration of what is actually driving this phenotype.

Our mutant-based evidence reveals a role for MEC-10 in the behavioral response to harsh touch, which is correlated with altered protein localization within the dendritic arbor. In addition, decrease in DEGT-1 and MEC-10 is correlated with morphological alternation in the PVD. Such morphological changes resemble isolation. In parallel, optogenetic activation of the PVD, which bypasses these channels, is also experience-dependent. The possibility of several layers contributing to the phenomenon indeed complicates our model, however altered protein localization may account for the pre-synaptic aspect. The downstream mechanisms were not explored in this paper, as the main focus of this work was placed on experience-dependent adult structural plasticity at the dendritic level.

•On line 398, the authors state "In summary, genetic, pharmacological and optogenetic evidence show that dendritic structural plasticity is an autonomous activity-dependent homeostatic mechanism. The mechanosensory dendritic tree morphology is independent from pre- or post-synaptic degenerin-mediated processes that affect behavioral escape in response to harsh touch." However, the optogenetic experiments do not support the conclusion that it is an activity dependent process as optogenetic activation in isolated animals across development could not rescue PVD branching abnormalities.

Indeed, we failed to recapitulate the crowded state by external optogenetic activation, however our system was not set for precise activation over an extended period, and as such may have provided insufficient stimuli. Additionally, optogenetic stimulation bypasses the degenerin mediated signal transduction and as such may fail to activate intracellular components which affect dendritic branching. We agree that this statement should have been phrased more clearly, and changed it accordingly (Lines 403-407).

•(441) While the authors did not find a correlation between PVD structure and animal response to harsh touch, was there a correlation between PVD structure and gait/crawling abnormalities? Tao et al. (2019) found that proprioception causes localized calcium events in the PVD dendrites, while harsh touch causes global PVD calcium events. Could ectopic PVD branches represent abnormal proprioception caused by the lack of external mechanosensory cues (therefore explaining the lack of correlation between branching and harsh touch avoidance behavior)? The authors state this as a possibility on line 334, but do not explore further.

As explained above in answer to reviewer #2 (point 4), we attempted to correlate the branching architecture of the PVD with crawling parameters, by analyzing the movement of isolated worms and then imaging and blindly quantifying the PVD structure. We found that ectopic dendritic branching densities of orders 2,3 and 4 failed to correlate with averaged movement parameters such as amplitude. Such a correlation, if present, is likely to be extremely noisy to detect, due to variability in both structure and movement. At the moment, our preliminary results fail to indicate any correlation between the various branching orders and crawling parameters on the singleanimal level.

-Are the suggested experiments realistic in terms of time and resources? It would help if you could add an estimated cost and time investment for substantial experiments.•The authors should have all other resources and strains required for the additional experiments. Additional mutant strains should be available from the CGC.-Are the data and the methods presented in such a way that they can be reproduced? Are the experiments adequately replicated and statistical analysis adequate?•For Figure S2A, the differences between the WT crowded and isolated, and mec-10 crowded and isolated animals are very similar, however the difference for WT is considered significant while the difference for mec-10 is not. The difference in the number of animals tested between the WT and mec-10 conditions is very large (WT crowded = 106, WT isolated=162, mec-10 crowded=29, mec-10 isolated n=14). Could there just not be a large enough n to make the difference between the mec-10 crowded/isolated condition significant?

Following reviewer’s comments we decided to exclude the proprioceptive data, and focus on our main findings describing plasticity at the PVD’s structural level and its behavioral response to harsh touch.

•There are some data not included that should be present. Specially, why is the data for the optogenetic stimulation rescue experiment not shown as a supplemental figure (mentioned on line 386)?

Thank for the suggestion, we have now added a new supplementary figure (Figure S11) where we show no effect of optogenetic stimulation on isolated worms.

Minor comments:Specific experimental issues that are easily addressable.•(178) When testing whether isolating animals after 24 hours of development, did the authors also transfer the control animals? Does harsh mechanosensory stimulation (induced through picking) affect the animals sensitivity to harsh touch when tested 24 hours later? It's interesting that in figure 1G, a higher (albeit, not significant) percentage of worms isolated after 24hrs respond to touch compared to crowded control. Could transferring underlie this increase in response

In the 24 hours isolation experiments the control animals remained in the plate, while the isolated animals were removed using an eyelash. This handling is presumably more similar to a gentle-touch event, and should not elicit a nociceptive response. The increase in touch response rates is rather interesting, however it seems to us unlikely that this singular gentle touch contact will trigger long-lasting effects, stronger than the constitutive collisions in the crowded state. In our calibration of the harsh touch assay, we have experimented with repeated contacts which ultimately elicit a habituation of the response; in our hands this habituation is reverted back to the initial response rate within several hours, such that even if the handling itself elicited a complete neuron habituation, this should not have lasting effects for the harsh touch assay itself. Nonetheless, we thank the reviewer for this observation, although at the moment we have no clear explanation for this interesting trend in the results.

•(386) Authors attempted to determine whether stimulating PVD in isolated worms could rescue morphological differences. They authors stimulated worms for 60s every 5 minutes for a 4 hour period. Does this optogenetic paradigm accurately replicate the experience of a worm on a crowded plate or is it too little stimulation? The authors did not include data on any other stimulus interval or duration.

The optogenetic stimulation paradigm was designed to elicit a robust activation for several hours, however we did not look into different illumination settings. Long term stimulation with light can potentially induce phototoxic damage. Based on our experiments involving animal isolation we determined that four hours are sufficient to induce some structural plasticity changes on the PVD, and the stimulation protocol was roughly set to try and elicit as many responses as possible but avoid constant stimulation which may elicit a habituation (for comparison, our harsh touch assay is a short contact followed by 10 second break).

Unfortunately, our system is not set for precise activation over an extended period, as it cannot track individual freely moving worms, and as such may indeed have provided insufficient stimuli. We have shown that the mechanosensory channels, degenerins, mediate the structural modifications on the PVD. The optogenetic stimuli induce current flow into the neuron while bypassing the native signals mediated by degenerins. This observation suggests that cation induced currents by light-gated channelrhodopsin are not sufficient for structural modifications of the PVD.

(127) Why do crowded osm-6 mutants have a lower percent of animals that respond compared to crowded N2? Typically, 80% of N2 crowded animals responded to harsh touch whereas only 35-40% of osm-6 crowded mutants responded. The authors do state that osm-6 mutants also have an abnormal PDE neuron (mechanosensory), however do not explicitly state whether this difference in the crowded response is due to impaired PDE function. Are there any PVD branching abnormalities in the osm-6 crowded mutants that could further explain this difference?

We did not test the structure of the PVD for *osm-6* mutants, as we were interested in the experience-dependency of the behavioral response. As the difference between crowded and isolated was robust, we did not examine PVD structure alterations. Both PVD and PDE mediate the response to harsh touch, and the lack of PDE should contribute a 50% reduction in the behavioral escape response, as previously published (Figure 3C published by Li et al., 2011). Our results support this observation.

•(157) Do the authors have a hypothesis as to why there is such a large difference in the percent of responding animals between the asic-1 and mec-10 crowded mutants? The authors present the data, but do not elaborate at all as to the underlying mechanisms other than that different amiloride-sensitive epithelial sodium channels may have positive and negative effects on harsh touch response. Is there a point in testing whether adding amiloride to the individual epithelial sodium channel mutants (including asic-1 and mec-10) in figure 1E to investigate which channels have positive/negative effects on harsh touch?

Different degenerin mutants do display very different responses under crowded and/ or isolated conditions, and the reviewer is correct in pointing the seemingly opposite effects of MEC-10 and ASIC-1. In our attempts to investigate which channels elicit which effect on harsh touch, we crossed and analyzed double and triple mutant combinations (see Figure S8), however the results we obtained were difficult to align in a classical epistasis analysis for the different structural and behavioral outputs. A global inhibition of degenerins under a single- or even triple-mutant background may mask some of the combinatorial effects rather than expose them. As such, we did not pursue this direction further. Future research may isolate novel combinations of mutants which will establish a genetic pathway; this indeed may be further probed by using amiloride to confirm the effect is directly related to amiloride-sensitive channels.

•(Figure S2) the authors determine whether there are significant differences between the crowded/isolated group for each genotype but do not determine whether there are significant differences across each genotype for the crowded versus isolated conditions. For crawling speed, while there is no difference between isolated and crowded conditions for WT and for mec-10 mutants, however there seems to be a significant difference between crowded WT and crowded mec-10 mutants. Further, mec-3 mutants (harsh-touch insensitive) show a significant isolation-induced difference in crawling speed which does not occur in WT or any of the other mutant strains tested. This finding is not explained nor further explored in the manuscript.

We thank the reviewer for the comment. For several considerations, we have decided to remove the proprioception results from this manuscript. We agree that the decreased speed seen in some backgrounds is interesting, however seems to be independent of other crawling parameters, that is, animals can retain the same crawling speed but alter crawling wavelength and amplitude.

•(338) The sentence needs to be changed to "…suggesting that the escape response…"

The sentence has been changed accordingly: "… suggesting that the escape response is not dependent on the structure of the dendritic trees, but on unknown downstream pathways”.

•There is an additional bracket in the citation on line 174.

This has been corrected accordingly.

Are the text and figures clear and accurate?•(175/198) Why are him-5 mutants used in the PVD mec-10 rescue strain and why are him5 mutants used to examine PVD branching? There is no justification in the text other than the figure caption which states that him-5 mutants were used as wild-type background for several strains after crossing. More justification is needed.

We used *him-5* mutants in order to cross the different double and triple mutants of the degenerin family for this project, but also in order to establish the reagents for a future project that focuses on PVD arborization in males. Having compared *him-5* and wild-type backgrounds, we have established that *him-5* worms show the same plasticity-induced changes at both the structural and the behavioral levels.

•Authors should add units to Figures S2A-C.

This figure was removed.

Do you have suggestions that would help the authors improve the presentation of their data and conclusions?•For the bar graphs, in addition to reporting the mean and standard deviation of each group, the individual response of each worm assessed could be indicated by a point.

The quantification of the harsh touch response has a binary nature. We believe that presenting dots for a binary quantification of ~40 animals will be very confusing and unnecessary. The percentage of responding worms, together with the total number of worms in the figure legend, should be sufficient to represent the data, and is clearer to understand in our opinion.

•In many of the PVD branching images, red and green are both used. The authors may want to consider using a colour-blind friendly colour combination (blue instead of red or green).

Thanks for the suggestion. We now use cyan instead of red.

Reference:

W. Li, L. Kang, B. J. Piggott, Z. Feng, X. Z. Xu, The neural circuits and sensory channels mediating harsh touch sensation in *Caenorhabditis elegans*. *Nat Commun* 2, 315 (2011).

Significance-Describe the nature and significance of the advance (e.g. conceptual, technical, clinical) for the field.•The authors conducted a genetic screen among degenerin channels and other receptors expressed in PVD to identify what genes affect isolation-induced changes in harsh touch response. Their work offers insight into how the different degenerin channels are involved in mediating changes in dendritic morphology in response to sensory experience. Specifically, the authors show that degenerin channels can affect behavior differently depending on sensory experience.•The authors show how isolation can cause changes in the dendritic structure of the PVD neuron in a time-resolved manner.-Place the work in the context of the existing literature (provide references, where appropriate).•Rose et al. (2005) showed that isolation during development can cause changes in the *C. elegans* gentle touch response and affects the connectivity/activity of underlying circuitry.-State what audience might be interested in and influenced by the reported findings.•This manuscript may be on interest to those in the fields of developmental biology, behavioral neuroscience, and those who use the *C. elegans* model system. In addition, this paper may be of interest to those who study mechanosensation or degenerins channels in different model organisms.-Define your field of expertise with a few keywords to help the authors contextualize your point of view. Indicate if there are any parts of the paper that you do not have sufficient expertise to evaluate.•Expert in how experience can impact behavior and function of the nervous system•Experience analysing *C. elegans* locomotor and behavioral phenotypes•Experience using optogenetics in *C. elegans*

[Editors’ note: what follows is the authors’ response to the second round of review.]

Based on the previous reviews and the revisions, the manuscript has been improved but there are some remaining issues that need to be addressed, as outlined below:1. There is consensus among the reviewers that Calcium imaging is needed to support the conclusions of this work.2. The authors are also encouraged to examine pre- and post-synaptic markers in PVD upon sensory deprivation.3. In your response letter, please address the reviewer comments listed below.Reviewer commentsReviewer 1: I remain uneasy about the adult contributions to the phenotypes seen, and the fact that siblings don't seem to have the effect- it makes me wonder what it is that is actually being studied. I also agree that calcium imaging could be done through collaboration, and would add quite a bit. The main question in my mind is whether they have demonstrated that the effects are due to mechanosensory stimulation.

We thank the reviewer for spelling out this central question. The mechanosensory functions of degenerins in *C. elegans* have been well described. The simple principle in this paradigm, is to generate mechanosensory-enriched environment, from both the sibling and the parents (see Materials and methods for more details). The approach of studying isolated versus crowded conditions was originally developed by the Rankin lab and we have adapted it for PVD structure and function. In our experimental design, the adults in the plates are indeed part of the mechanosensory environment, but they are only a minority in a plate enriched with their progeny. The source of mechanosensory stimulation is composed of the high frequency, random and non-random, mechanosensory interactions within the tested group of siblings and between the sibling and the adult worms.

Altogether, our results support mechanosensation as the main plasticity inducing modality in this assay. We consider mechanical stimulation to be the main contributor for this effect, since glass beads provide some level of mechanical stimuli as to partially rescue the isolated phenotype (Figure 1—figure supplement 1A) whereas chemical stimulation does not (Figure 1D). Additionally, chemosensory mutants still show isolation-dependent behavioral and dendritic plasticity (Figure 1—figure supplement 1B). This conclusion is further supported by the cell-autonomous role of mec-10 in PVD mechanosensory perception, responsible both for loss of nociception in crowded conditions (Figure 1E, F) and for PVD morphological defects (Figure 2B-E).

In our answer to reviewer #1 from 08.2022, we showed that we do get a dosage effect on the morphological isolated-like phenotypes; when we use 250 eggs, instead of one isolated egg, we get some morphological phenotypes suggesting that the amount of mechanical stimulation is important. See answer to reviewer #3 for more on these controls.

Thank you for raising the idea for calcium imaging, which we pursued by collaboration following your suggestion. We performed two different sets of calcium imaging experiments, comparing crowded to isolated worms. The first set of experiments was set to determine whether sensory experience alter the baseline calcium dynamics, while a second set observed the response of PVD to direct mechanosensory stimulation. Our analysis did not reveal any change in PVD calcium activity between isolated and crowded animals, neither in baseline nor in response to mechanical stimuli.

These results suggest that the morphological changes in the PVD are not directly related to the calcium dynamics in the cell body of the PVD, and that the plasticity in response to harsh touch is mediated by circuit mechanisms that act downstream to the cell body of the PVD.

Reviewer 2: The authors have responded to all the queries made by the reviewers. It is understandable that sensory experience-dependent behavioral plasticity and dendrite arborization is not related. That is also suggested by another manuscript (Tao et al. 2019 Dev Cell) that dendrite structure is related to body posture, whereas harsh touch is mediated by the synaptic connection. PVD acts as a polymodal sensory neuron.The authors need to discuss the point clearly that the changes in the dendrite structure caused by the sensory deprivation might be very small to cause any quantitative changes in the body posture. As there is not a great assay developed for proprioception in worm.

We thank the reviewer for this comment. Indeed, we find no evidence for a causal link between PVD morphology (structure) and nociception (function), in the current study or elsewhere. This fact can be explained by synaptic differences, rather than the morphological contribution of the PVD dendritic tree. Our model is that the downstream synaptic targets affect the behavioral output, and we have revised the manuscript to state this in a clearer way (Discussion, lines 542-545).

We also explicitly discuss the work of Tao et al., 2019 and the point that changes in the dendrite structure caused by the sensory deprivation might be very small to cause any quantitative changes in the body posture (See Discussion lines 515-520). In a previous version of this manuscript (Inberg et al., 2021; doi: https://doi.org/10.1101/436758), we showed that isolation affects the crawling gait by increasing the amplitude and wavelength independently of the gentle touch circuits. Based on previous reviews we removed that section to make the manuscript more focused and readable.

However, I noticed that in the existing contexts discussed in the paper, the authors could address the experimental suggestion made by the referees. Especially, pre or post-synaptic changes in PVD neurons upon sensory deprivation. or Calcium dynamics in PVD or/and in the neuron postsynaptic to PVD.This perhaps could improve the manuscript.

Thanks for this suggestion, which we indeed followed through collaboration. Following two sets of calcium imaging experiments performed on the PVD cell body we found no significant differences in calcium dynamics between animals grown under isolated and crowded conditions (see Figure 6; Figure 6—figure supplement 1, 2; Videos 5-7) and answer to Reviewer #(1). While revising the text, we now also elaborate more on the subject of pre- and post- synaptic plasticity, also in light of the calcium imaging experiments (see lines 453-459; 506-509; 566-571). Since this work focuses on mechanosensory effects taking place at the sensory level itself (PVD and its dendritic arbor morphology), and considering that PVD synapses onto an intricate sensory circuit (Husson et al., 2012), we believe that in order to keep the manuscript concise such downstream effects would be the focus of a future, follow-up study.

Reviewer 3: The authors have satisfactorily addressed many of the reviewer's comments. In the response to reviewers, the authors defended the lack of a clear mechanism by indicating that this manuscript focused on describing the behavioral and morphological phenomena of isolation and showing the involvement of two genes; however, because the behavioral and the anatomical changes do not correlate, this negatively impacts the overall quality of this manuscript. It could be particularly informative to examine changes in pre- and post-synaptic markers and/or the calcium dynamics in the PVD neurons, given that their optogenetic data suggest the behavioral effects of isolation appear to be post-sensory, and these experiments could potentially answer the question of whether the behavioral changes are mediated by changes in cell excitability or synaptic strength. There are a fair number of labs that have performed calcium imaging experiments on PVD who might be willing to collaborate- this could be a potential avenue of the investigation of the mechanisms. Because the 2 measures do not correlate the paper is not as impactful as it would be if there was some understanding of why they did not.

Thank you for these suggestions. Following this and comments by other reviewers, we examined the response of PVD using calcium imaging experiments, which we indeed performed through collaborations (see also answers to Reviewers #1 and 2). We performed two different sets of calcium imaging experiments, comparing crowded to isolated worms. One set of experiments asked whether baseline calcium dynamics would differ between crowded and isolated animals, while the second set probed the response of the PVD to a controlled set of external mechanosensory stimulations.

Both for baseline and mechanical stimulation assays, the PVD’s calcium response appeared similar for crowded and isolated animals. We have now added these results to the revised manuscript. As described in the results and discussion we also unexpectedly found that mechanosensory stimulation of the PVD results in an increase in calcium levels not only in the ON phase of the stimulation, as described before, but also on the OFF phase, when the stimulus is removed.

A new issue is that in the response to reviewer #1, the data shown in Figure 1 is problematic. The reviewer asked whether there were behavioral differences in the populations of worms reared in different conditions, however, Figure 1 described the dendritic structural differences, and no behavioral data was presented. Additionally, data in Figure 1 appear to suggest that some effects of crowding are dependent on the parents (1AandB), while others are dependent on the siblings (1C) – it is important to see the behavioral effects in these same populations and some discussion of these observations.

Thank you for the comment. Indeed, in the response for reviewer #1 from August 2022, we provided evidence in control experiments showing that even starting with 250 eggs, instead of a single isolated egg, we observe structural changes in the PVD arbors that are similar between them and significantly different from the crowded worms. This is true for two out of three morphological parameters. We do not expect that the harsh touch assay or posture will be sensitive enough to detect differences between the dosage of worms in the plate. See also answer to reviewer #1. The results we show in the manuscript are uniformly achieved by crowding the plate with the progeny of 30 adult worms. While these worms remain on the plate, we consider the predominant mechanosensory ‘experience’ as the contacts between the siblings of these 30 adults.

We decided not to include the control Figure 1 from answers to reviewers (30.08.2022) so as not to complicate the text further, and we believe the other controls we discuss remain sufficient to support the importance of mechanosensory stimulation in our assays.

As we show in Figures 1 and 5I, the response to harsh touch and to optogenetic stimulation of the PVD are significantly lower for isolated worms. Taken together with the calcium dynamics results, we suggest that the behavioral plasticity involves mechanisms downstream to the PVD, and is independent of the PVD’s intrinsic calcium activity and structure.